# Denominal *-ed* Adjectives and Their Adjectival Status in English Morphology

**Takashi Ishida**

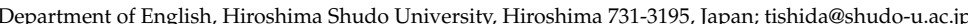

Department of English, Hiroshima Shudo University, Hiroshima 731-3195, Japan; tishida@shudo-u.ac.jp

**Abstract:** In the study of English denominal adjectives, scholarly attention has predominantly centred on those with Latinate suffixes (e.g., *-al*, *-ary*, and *-ic*/*-ical*), which are well-known as relational adjectives (RAdjs) and are extensively scrutinised in the existing literature. Conversely, those with English native suffixes (e.g., *-en*, *-ern*, *-y*, and *-ed*) have not undergone thorough examination to date. In the present study, I delve specifically into denominal adjectives with the suffix *-ed* (*-ed* Adjs), such as *bearded*, *long-tailed*, and *shirt-sleeved*. I present a novel basic picture of these adjectives, setting forth the following two central propositions: (i) *-ed* Adjs are a type of RAdj and (ii) undergo conversion to qualitative adjectives (QAdjs) (e.g., *bearded man* vs. *bearded rock*) akin to the better-known Latinate RAdjs (e.g., *grammatical error* vs. *grammatical sentence*). The analysis is conducted by examining suffixal etymology (i.e., Latinate or Germanic), suffixal properties (i.e., *all-purpose* or *dedicated*), and the driving factor for QAdj-forming conversion (i.e., the modal attribute *true*). These propositions and analyses collectively enrich our comprehensive understanding of the semantic and morphosyntactic properties of *-ed* Adjs within the realm of English morphology.

**Keywords:** adjectivalisation; relational adjective; proprietive semantics; suffixal nature; secondary word-class conversion

## 1. Introduction

In English derivational morphology, as clearly articulated by Bauer et al. (2013, at the very beginning of Ch. 14), Fábregas and Marín (2017), and Sleeman (2021, p. 551), studies on adjectivalisation remain comparatively scarce compared to those on verbalisation (e.g., Clark and Clark 1979; Hale and Keyser 1993; Levin and Rappaport Hovav 1994; Goldberg 1995; Plag 1999; Borer 2005) and nominalisation (e.g., Chomsky 1970; Grimshaw 1990; Alexiadou 1998; Harley 2009; Lieber 2016).

On the other hand, there has been a gradual increase in scholarly efforts focussing on adjectivalisation, particularly within the domain of denominal adjectives, commencing with seminal works by Ljung (1970), Levi (1978), and Beard (1995). Notable contributions have emerged from scholars such as Giegerich (2005, 2009), Fábregas (2007, 2020), Fradin (2008, 2017), Bisetto (2010), Cetnarowska (2013), Rainer (2013), Spencer (2013), Spencer and Nikolaeva (2017), and Nikolaeva and Spencer (2013, 2020), as well as, significantly, a comprehensive series of works by Nagano (2013, 2015, 2016, 2018a, 2018b), Nagano and Shimada (2015), Shimada and Nagano (2018), and Nagano (2023) as the latest contribution. Thanks to these remarkable previous studies, the properties of denominal adjectives have been systematically and deliberately advanced, with cross-linguistic implications.

However, the majority of the existing literature on adjectivalisation, with the exception of Nagano (2023), has generally specialised in denominal adjectives with typical non-native (i.e., Latinate) suffixes such as *-al*, *-ary*, or *-ic*/*-ical* (e.g., Levi 1978; Kaunisto 2007; Bauer et al. 2013, p. 318; Plag 2018, pp. 95–96). In this paper, I specifically address adjectives suffixed with the native *-ed*. While the suffix *-ed* remains considerably productive in English, especially when compared to other native suffixes that are no longer productive, such as *-en* and *-ern* (Bauer et al. 2013, pp. 304–6), there are limited systematic studies on this suffix

and its derivatives due to the inherently complex and multifunctional nature of *-ed* as a homophonous affix (e.g., *-ed* appears in various grammatical and morphological contexts, including passive constructions, verbal past tense, verbal/adjectival participles, and the derivation of adjectives). The present study explores denominal adjectives suffixed with *-ed*, such as *bearded*, *long-tailed*, and *shirt-sleeved*, hereinafter referred to as '*-ed* Adjs'.

The immediate concerns surrounding *-ed* Adjs have primarily revolved around their semantico-pragmatic aspects. For example, Hudson (1975) observes that constraints exist concerning the acceptability of phrases containing these adjectives, such as *a bearded/*wifed man*, *an eyed *boy/probe*, *a verandahed/*doored bungalow*, and *a *large-gardened/red-roofed house*.[1] Researchers have rigorously explored these constraints, employing key notions such as *inalienable possession* (i.e., a close, inherent relationship between two entities where the possessee cannot be conceptually or physically separated from its possessor) and *informativity* (i.e., the degree to which a given piece of information or linguistic expression provides new, relevant, or useful knowledge to the recipient) (e.g., Hirtle 1969; Hudson 1975; Beard 1976; Ljung 1976; Quirk et al. 1985; Goldberg and Ackerman 2001; Nevins and Myler 2014; Ishida 2023; Morita 2024). While semantico-pragmatic constraints have been discussed from many angles, the morphosyntactic properties and adjectival status of the adjectives in question have not yet undergone adequate investigation and clarification. Against this backdrop, the questions that remain to be solved are threefold:

- What specific properties do *-ed* Adjs intrinsically possess?
- Into which adjectival categories should *-ed* Adjs be classified?
- If *-ed* Adjs are primarily relational adjectives, how can their behaviours as qualitative adjectives be analysed and understood?

The aim of this study is to provide plausible answers to these inquiries and to offer a holistic view of the derived adjectives, exploring and unveiling their adjectival status. Specifically, the contention posits that *-ed* Adjs can be legitimately categorised as a type of relational adjective. This theory can be reasonably deduced through a discussion of conversion.

The examination unfolds in the following manner. Section 2 first provides an overview of the canonical properties of the following two distinct categories of adjectives: relational adjectives (hereafter RAdjs) and qualitative adjectives (hereafter QAdjs). This enables us to form a general framework of the domain of denominal adjectives and allows us to understand the subject of analysis, i.e., *-ed* Adjs. Section 3 provides an overview of the basic (albeit irregular and idiosyncratic) properties of *-ed* Adjs, drawing particular attention to their phonology, semantics, and morphosyntax. Section 4 examines the adjectival status of *-ed* Adjs, revealing their intrinsic nature as RAdjs, and further discusses why we should assume the adjectives to be a type of RAdj, taking into account the suffix's etymology and properties (e.g., Rainer 2013). Section 5, predicated on Nagano's (2018a) conversion analysis, elucidates the enigma and rationale behind the idiosyncratic behaviour exhibited by *-ed* Adjs from the perspective of proprietive–similative polysemy. Finally, Section 6 concludes the paper with some summarising remarks.

## 2. Two Types of Denominal Adjectives and Their Canonical Properties

Relational adjectives (RAdjs), also known as associative adjectives, constitute a subset of denominal adjectival formations wherein the role of the adjectivalising morpheme is solely transpositional (e.g., *senatorial*, *polar*, *algebraic*). Morphosyntactically, they function as adjectives by directly modifying nouns and agreeing with the modified noun in languages featuring agreement morphology. Semantically, RAdjs *classify* the noun that they modify by establishing various semantic relationships between their base nouns and the modified nouns (Giegerich 2005, 2009; see also Shimamura 2014; Nagano 2016). For example, *algebraic mind* means 'a mind having to do with algebra, referring to algebra, characterised by algebra' (Plag 2003, p. 94; Nagano 2016, p. 43). It is also widely recognised that, due to their exclusively transpositional nature, RAdjs exhibit "exactly the same range of denotations" (Nikolaeva and Spencer 2020, p. 92) as their base nouns. Therefore, RAdj-N(oun) and N-N

pairs are "fully synonymous and, more interestingly, are apparently used in free variation" (Levi 1978, p. 224), as demonstrated in (1) (examples are adopted from Levi 1978, p. 38; see also Beard 1995, p. 188; Levi 1978, p. 224).[2]

| (1) | a. | atomic bomb | ≈ | atom bomb |
|---|---|---|---|---|
| | b. | maternal role | ≈ | mother role |
| | c. | industrial output | ≈ | industry output |
| | d. | marine life | ≈ | ocean life |
| | e. | linguistic skills | ≈ | language skills |
| | f. | urban parks | ≈ | city parks |

The relational meaning that combines the base noun of the adjective and the head noun is regarded as extra-linguistic, closely tied to the conceptual knowledge associated with the two nominal concepts (Rainer 2013, p. 15; see also Nikolaeva and Spencer 2020, pp. 91–95).

In contrast, qualitative adjectives (QAdjs), also referred to as ascriptive adjectives, serve to *qualify* the noun from which they are predicated (e.g., *beautiful*, *picturesque*, *nervous*). To succinctly illustrate these two distinct types of adjectives, compare *grammatical* in (2), where (2a) is cited from Plag (2003, p. 94, italics mine) while (2b) is from Nagano (2018b, p. 276).

| (2) | a. | a *grammatical* sentence |
|---|---|---|
| | b. | a *grammatical* error |

The adjectives in (2) have the same form, yet they exhibit entirely different meanings and functions in modifying each head noun. *Grammatical* in (2a) is considered a QAdj, as it means 'conforming to the rules of grammar' and is predicated on the attribute of *sentence* (i.e., *the sentence is grammatical*). In contrast, *grammatical* in (2b) is an RAdj, as it has a relational sense such as 'having to do with grammar' and does not predicate the characteristic of *error* (i.e., \**the error is (very) grammatical*) (Nagano 2018b, p. 276) but rather subcategorises the type of *error* (i.e., *an error of grammar*). Thus, whereas QAdjs have a qualifying/predicating function, RAdjs have a classifying function.

In addition to their relational semantics and classificatory function above, the other widely assumed prototypical characteristics of RAdjs can be summarised in (3), as detailed by scholars such as Levi (1975, 1978), Beard (1995), Giegerich (2005, 2009), Bisetto (2010), Cetnarowska (2013), Shimamura (2014), and Nagano (2015, 2016, 2018a).

| (3) | a. | Predicability: | \**this output is industrial*, \**this decision is senatorial* |
|---|---|---|---|
| | b. | Gradability: | \**a very industrial output* |
| | c. | Comparability: | \**more industrial* |
| | d. | Coordinability: | \**the big and wooden table* |
| | e. | Quantifiability: | *monochromatic, binational, triconsonantal, multiracial* |
| | f. | Nominalisability: | ??*presidentialness*, ??*racialness* |

Regarding (3a), RAdjs cannot be used in predicative constructions. For (3b) and (3c), RAdjs do not exhibit gradability or comparativeness. Concerning (3d), RAdjs cannot be coordinated with QAdjs. As for (3e), RAdjs can be affixed by numerical prefixes. Lastly, as in (3f), RAdjs do not allow for further nominal affixation. Incidentally, as observed by Farsi (1968), the choice of negative prefix can also differentiate one word from another, such as *nongrammatical* (RAdj) vs. *ungrammatical* (QAdj) (see Farsi 1968, p. 53 for other contrastive pairs).

The behaviour of RAdjs in (3) should be derived from their base nouns, as nouns generally have these properties. For example, when the head noun is a deverbal noun, the base noun of an RAdj is interpreted as its internal/external argument with specific prepositions, such as *of* and *by*, as shown in (4), cited from Nagano (2015, p. 5).

| (4) | a. | *Japanese* democratisation after World War II |
|---|---|---|
| | | = the democratisation *of Japan* after World War II |
| | b. | the *Japanese* attack on Pearl Harbour |
| | | = the attack on Pearl Harbour *by Japan* |

Building on this, Nagano (2013, 2015) extends Beard's (1995) argument and analyses RAdjs as alternative forms of prepositional phrases (hereafter PPs), as indicated in (5) and (6), excerpted from Nagano (2013, p. 123; 2015, pp. 6–7; see also Fradin 2008, §4).

| (5) | | | | | |
|---|---|---|---|---|---|
| | a. | *presidential* plane | a′. | plane *of the president* | |
| | b. | *cellular* structure | b′. | structure *of cells* | |
| | c. | *dental* disease | c′. | disease *of teeth* | |
| | d. | *bearded* man[3] | d′. | man *with a beard* | |
| | e. | *southern* exposure | e′. | exposure *to the south* | |

| (6) | | | | | |
|---|---|---|---|---|---|
| | a. | *preadverbial* expression | a′. | expression *in front of an adverb* | |
| | b. | *postnominal* adjective | b′. | adjective *after a noun* | |
| | c. | *suprasegmental* phonemes | c′. | phonemes *above segments* | |
| | d. | *sub-Saharan* Africa | d′. | Africa *below the Sahara* | |
| | e. | a *trans-global* expedition | e′. | expedition *across the globe* | |

Interestingly, even if the PPs are complex, such as those in (6), the corresponding RAdjs similarly exhibit the same nominal properties, as aforementioned in (3). As further emphasised by Nagano (2015, p. 7), the observations so far lead us to understand that there is a morphological parallelism between RAdjs and PPs; namely, while nonprefixed RAdjs correspond to simple PPs, as in (5), prefixed RAdjs conform to complex PPs, as in (6).

## 3. Basic Properties of *-ed* Adjectives

This section provides an overview of the basic properties of *-ed* Adjs, focussing on their (i) input structure, (ii) phonology, (iii) semantics, and (iv) morphosyntax. Although this study cannot examine and discuss each property of *-ed* Adjs in depth, it is crucial to identify and shed light on the issues involved. The semantic and morphosyntactic properties and problems particularly relevant to the present study are summarised at the end of the section (Section 3.5).

### 3.1. Input Structure

Let us first survey the conceivable structures that can serve as the input for *-ed* Adjs. Before proceeding, it is important to note that the examples are primarily cited from Marchand (1969, pp. 264–67), Meys (1975, pp. 163–64), Allen (1978), Gram-Andersen (1992), Bauer et al. (2013, p. 304), and COBUILD (2017, p. 94) (see their respective works for other various examples), in addition to the *Oxford English Dictionary Online* (henceforth OED 2023). To the best of my knowledge, the cited data from OED (2023) remain synchronically in use.

Regarding the typical morphological characteristics of *-ed* Adjs, Bauer and Huddleston (2002, p. 1709) assert that "[t]he construction is extremely productive" and OED (2023, s.v. *-ed suffix₂*) goes as far as to say that the adjectivalising suffix *-ed* can be attached "without restriction to any noun" (see also Bauer et al. 2013, p. 306; Plag 2018, pp. 150–51). This statement, however, may appear too strong, particularly when considering the unacceptable cases mentioned in the Introduction (e.g., *an eyed boy; *a carred lady; *a wifed man*). Nevertheless, it is true that this suffix can be affixed to various kinds of bare noun in (7) and even more complex elements, as enumerated in (8) to (10).

| (7) | | Bare noun: |
|---|---|---|
| | a. | bearded, toothed, freckled, tongued, voiced, skinned, throated, toed, heeled |
| | b. | booted, braceleted, jacketed, armoured, barbed, gloved, hooded, turbaned |
| | c. | cultured, fashioned, mannered, principled, skilled, diseased, jaundiced |
| | d. | patterned, pointed, spotted, dotted, striped, beaded, sized, styled, curved |
| | e. | wooded, pebbled, treed, tufted, flowered, leaved, mossed, hued, branched |

(8)    Noun phrase (NP):[4]
    a.    blue-eyed, red-faced, long-legged, four-footed white-haired, heavy-handed
    b.    heavy-booted, white-gloved, double-buttoned, black-coated, high-crowned
    c.    broad-minded, low-spirited, soft-mannered, rough-featured, crazy-minded
    d.    one-sided, three-cornered, double-edged, multi-coloured, middle-sized
    e.    muddy-rivered, sandy-beached, deep-rooted, thick-stemmed, three-awned

(9)    Compound noun:
    a.    swallow-tailed, honey-tongued, pigeon-toed, pig-headed, lion-hearted
    b.    shirt-sleeved, bowler-hatted, rubber-soled, leather-aproned, sun-hatted
    c.    crime-minded, party-spirited, soul-diseased, grit-tempered, world-minded
    d.    pencil-pointed, hill-sided, cone-shaped, heart-shaped, razor-edged
    e.    beach-pebbled, oak-treed, garden-flowered, pine-forested, rain-weathered

(10)    Bound stem:
    a.    naked (legs)
    b.    wicked (uncle)
    c.    blackavised (chap)
    d.    jagged (edge)
    e.    fructed (tree)

The five divisions of input in (7) to (9), namely (a) to (e), can be roughly categorised as follows: (a) body parts; (b) wearable items such as clothes and accessories; (c) human properties or characteristics; (d) dimensions; and (e) natural objects. The remarkable productivity of the *-ed* suffix is evident.

The examples provided in (7) suggest that the suffix *-ed* can be attached to a wide range of bare nouns without strict limitations.[5] The input structure of *-ed* Adjs in (8) and (9), composed of sequences combining a two-element nominal base and *-ed*, is sometimes referred to as an extended bahuvrihi compound, as exocentric Adj-N and N-N compounds are occasionally equivalent to extended forms in *-ed*, such as *bareback*(*ed*), *duckbill*(*ed*), *long-nose*(*d*), *bighead*(*ed*), and *faint-heart*(*ed*) (Marchand 1969, pp. 265–66; Adams 2001, p. 94).[6]

Regarding the bound stems in (10), some of the stems are not found in OED (2023) and other dictionaries, which may suggest that they are borrowed or have obscure origins. Harley (2006, pp. 157–58) argues that the base of *naked* in (10a), for instance, has become obsolete or uncommon, asserting that "[t]he infrequency of the root in words like *naked*, *jagged*, and *wicked* means that it might not have occurred to you before that the *-ed* in these words is a suffix at all". This prompts speculation that the suffix may have undergone a process of reanalysis, amalgamating with a cranberry morpheme like *nake-*.[7] However, this assumption warrants careful examination.[8] Notably, the editors of this issue perspicaciously remark that these *-ed* Adjs can also be classified, albeit approximately, into the same semantic types. While *naked* in (10a) may appear challenging to classify, it has either or both semantic properties associated with *body parts* and *wearable items*.[9] The next two *-ed* Adjs, *wicked* in (10b) and *blackavised* in (10c), connote *human properties*. Lastly, *jagged* in (10d) denotes *dimensions*, while *fructed* in (10e) denotes *natural objects* (or potentially dimensional attributes as well).

### 3.2. Phonological Properties

In the examination of *-ed* Adjs, attention is consistently drawn to their phonological properties due to their intriguing peculiarities (e.g., Marchand 1969, p. 264; Allen 1978, p. 39; Fudge 1984, pp. 65–66).

(11)    a.    /t/:    after voiceless consonants                e.g., *shaped*, *forked*
    b.    /d/:    after voiced consonants or vowels        e.g., *diseased*, *cultured*
    c.    /ɪd/:    after dental sounds such as /t/ or /d/    e.g., *booted*, *bearded*

In comparison to the general phonological rules outlined in (11), it has been acknowledged that the suffix *-ed*, in certain words originating from Old and Middle English, is consistently pronounced, as illustrated in (11c). Examples include *crabbed*, *cragged*, *crooked*, *cussed*, *dogged*, *forked*, *jagged*, *knobbed*, *peaked*, *piked*, *ragged* (cf. *rugged* is the ablaut variant), *wicked*,

and *wretched* (Marchand 1969, p. 265). Allen (1978, pp. 39–40) explains that these words are "clearly historical relics, having a literary or picturesque flavour in most cases", such as *a peaked* /pikɪd/ *mountain* (contrasted with *high-peaked* /pikd/) and *a forked* /fɔrkɪd/ *tail* (contrasted with *three-forked* /fɔrkd/), and are no longer synchronically productive formations with *-ed* (Allen 1978, p. 40).

In addition to Marchand's denominal *-ed* examples, Fudge (1984, p. 66) observes that certain examples, including deverbal adjectives such as *accursed*, *beloved*, *blessed*, *cursed*, *naked*, and *sacred*, are also pronounced /ɪd/ even when not preceded by /t/ or /d/. Notably, some of them are variants of verbal past participle forms, as demonstrated in (12).

|  |  | X*ed* | /ɪd/ | /d/ (past form of 'to V') |
|---|---|---|---|---|
| (12) |  |  |  |  |
|  | a. | aged | 'old' | 'to age' |
|  | b. | dogged | 'persevering' | 'to dog' |
|  | c. | learned | 'erudite' | 'to learn' |
|  | d. | ragged | 'in rags' | 'to rag' |

For instance, *dogged* in (12b) means 'persevering' when pronounced with /ɪd/ (i.e., *dogged* as a denominal *-ed* Adj), whereas it means 'to dog, to follow someone closely and continuously' with /d/ (i.e., *dogged* as a past form of the verb *dog*).[10]

Similar phonological distinctions are observed between adjectival and verbal passives, i.e., in the deverbal domain. Dubinsky and Simango (1996, §6) scrutinise the syntactic distribution of two identical passive forms (adjectival and verbal) based on their phonologically distinct behaviour. They suggest that verbs such as *seem*, *remain*, *sound*, and *look* exclusively select adjectival complements with *-èd* (i.e., /əd/, /ɪd/), such as *abhorrèd*, *stripèd*, *avowèd*, *markèd*, *cursèd*, *believèd*, *hallowèd*, *redeemèd*, *despisèd*, and *unstoppèd*. Dubinsky and Simango (1996, p. 777–78) conclude, based on their English historical survey of metrical verse, that *-ed* represents two distinct affixes and merged (sometime between the seventeenth and nineteenth centuries) into the identical spelling *-ed* in modern English. Importantly, although the adjectival *-ed* /ɪd/ has disappeared in its full syllabic form, Dubinsky and Simango (1996, p. 779) note that there are speakers who are "still quite able to use it correctly and productively (in some dialects)". This possibility implies that, from a phonological perspective, there is still room to investigate a means of differentiating not only between the adjectival *-ed* and the verbal *-ed* in the deverbal domain, but also amongst the examples in (11) and (12), i.e., in the denominal domain.

### 3.3. Semantic Properties

While considering the semantic properties of *-ed* Adjs, the terminological ambiguity surrounding them should also be addressed and disambiguated. There exist several terms for *-ed* Adjs, including *possessive*, *possessional*, and *proprietive*. In accordance with the arguments presented by Nikolaeva and Spencer (2020, p. 101) and Nagano (2023, pp. 500–2), I adopt the term *proprietive* in this paper as a more precise semantic descriptor for *-ed* Adjs rather than the alternative terms.

Bauer et al. (2013, p. 313) analyse the suffix *-ed* as expressing an ornative sense, generating adjectives from nouns with the overarching meaning 'having X, being provided with X', wherein X corresponds to the base noun of the derivative.[11] Illustrative instances include *a bearded man* 'a man with a beard'; *a wooded area* 'an area covered with trees'; and *a blue-eyed boy* 'a boy having blue-eyes'. Here, the semantic relationship between the base noun of the derivative (e.g., *beard*) and the head noun (e.g., *man*) is the property−proprietor relationship. This semantic relationship can be illustrated by Nikolaeva and Spencer's (2020, p. 102) examples in (13), where the nouns attached by PROP.ADJ stand for potential derived adjectival forms of those nouns, accompanied by certain possible suffixes or case markers.

(13)    a.    campsite-PROP.ADJ riverbank
                'a riverbank which has (many) campsites along it'
        b.    riverbank-PROP.ADJ campsite
                'a campsite which is by a riverbank (as opposed to in the forest)'

In both examples in (13), albeit the constituents, *campsite* and *riverbank* are indeed identical, and their proprietive relationship diverges by virtue of the proprietive marker. Thus, whereas *campsite* in (13a) serves as a property for *riverbank* as the proprietor, this semantic relationship is patently reversed in (13b).

Despite previous studies traditionally categorising *-ed* Adjs as *possessive* adjectives (e.g., Marchand 1969, p. 264), this terminology appears to be misleading and, at times, obfuscating, as possessive adjectives basically refer to pronominal possessors (or possessive determiners), like *my/your/their* (*book*), and ethnic adjectives, such as *Galilean* (*revolution*) and *Jovian* (*moon*) (Nikolaeva and Spencer 2020, p. 96). Nikolaeva and Spencer (2020, p. 101) regard proprietive adjectives as being akin to possessive and relational adjectives, as they share the general concept of 'being possession' between the semantic content of the base noun and the head noun. However, the difference between proprietive and possessive functions should be attributed to the distinct nature of semantic headedness in noun phrases (Nikolaeva and Spencer 2020, p. 101; see also Beard 1995, pp. 219–27). With possessive adjectives (Poss.Adj), the relationship with the modified noun takes the form in (14a); conversely, with proprietive adjectives (Prop.Adj), the relationship is inverted, as in (14b).

(14)    a.    Poss.Adj N:    $\text{Poss.Adj}_{<possessor)>}\ \text{N}_{<possession>}$    e.g., *my book*
      b.    Prop.Adj N:    $\text{Prop.Adj}_{<possession>}\ \text{N}_{<possessor>}$    e.g., *bearded man*

Additionally, as observed by Akiko Nagano (personal communication), a similar reversal of the possession relationship can be found between the clitic and the *-ed* Adj, as compared in (15).

(15)    a.    the boy's blue-eye:    the $\text{boy's}_{<possessor>}$ blue $\text{eye}_{<possession>}$
      b.    a blue-eyed boy:    a $\text{blue-eyed}_{<possession>}\ \text{boy}_{<possessor>}$

In this respect, even though clitics are not adjectives, they can be grouped together with possessive adjectives in terms of the modification relationship between the modifiers and the head nouns, although not with proprietive adjectives like *-ed* Adjs.

On the other hand, another term that can be employed is *possessional* adjectives (e.g., Jespersen 1954, II, p. 375; Beard 1976, 1981a, 1981b, 1995), which generally indicates that a noun is in possession of (or exhibits a quality of) some other noun. This term could be considered a candidate to be applied to the class of *-ed* Adjs; however, in reality, it encompasses a broad range of adjectives with other rival suffixes, such as *warty*, *knowledgeable*, *intellectual*, *sorrowful*, *stylish*, *nodose*, *nodulous*, and *modular* (Beard 1976, p. 155; 1981a, p. 34; 1981b, p. 36; Szymanek 1996, p. 258). Hence, I regard *possessional* as a superordinate term for *-ed* Adjs and *proprietive* as a more specific term for them.

On the native suffixes deriving proprietive adjectives, Nagano (2023, p. 509) analyses the affixal rivalry between *-ed* and *-y* (e.g., *boned* vs. *bony*; *brained* vs. *brainy*; *edged* vs. *edgy*; *mooded* vs. *moody*; see Nagano 2023, pp. 506–7 for additional examples). Building on Kennedy and McNally's (2005) analysis of gradable adjectives and their four scale structures (i.e., (i) CLOSED SCALE; (ii) LOWER CLOSED SCALE; (iii) UPPER CLOSED SCALE; and (iv) OPEN SCALE (cf. Murphy 2010, §11.5.2)), she posits that, while proprietive *-y* adjectives (hereafter *-y* Adjs) represent the totally open-scale type, *-ed* Adjs manifest the closed-scale type. To exemplify this distinction, *-ed* and *-y* Adjs exhibit dissimilarities in their manner of modification with *very*, a marker associated with totally open-scale attributes, and can exclusively modify *-y* Adjs, as compared in (16) (examples taken from Nagano 2023, p. 507; cf. ibid., p. 510).

(16)    a.    ??very dressed      very dressy
      b.    ??very fished[12]      very fishy
      c.    ??very (left-)handed      very handy

Additionally, Nagano (2023, p. 509) highlights that *-ed* Adjs have a morphological antonym, such as *-less* (e.g., *clouded* vs. *cloudless*, *legged* vs. *legless*) (see also Nikolaeva and Spencer (2020, p. 105), for discussions of *caritive* (or *privative*) adjectives, meaning 'lacking-N' or

'without N' as negative counterparts of proprietive adjectives. They further consistently accept maximality modification via *completely*, as evidenced in phrases such as *completely cloudless*.[13]

In addition, Allen (1978, p. 245) explains that while the typical semantics articulated by the *-ed* suffix is 'characterised by the presence of X' and extends beyond a proprietive meaning (see also OED 2023, s.v. *-ed suffix₂* for the sense 'possessing, provided with, *characterised by*' (something), italics mine), certain derivatives express a simile-based meaning, as exemplified in (17), cited from Allen (1978, pp. 269–70; cf. ibid., p. 249).[14]

| | | | |
|---|---|---|---|
| (17) | a. | child-fingered | 'having fingers like a child' |
| | b. | eagle-eyed | 'having eyes like an eagle' |
| | c. | gorilla-armed | 'having arms like a gorilla' |
| | d. | lion-clawed | 'having claws like a lion' |
| | e. | grass-seeded | 'having seeds like grass' |
| | f. | butcher-boned | 'having bones like a butcher' |

This reading gains support from the observation that such *-ed* Adjs can be applied to entities beyond the denotation of the modifier. For instance, *eagle-eyed* in (17b) can describe things other than eagles, as illustrated in *an eagle-eyed third baseman*. This flexibility is similarly evident in phrases like *swallow-tailed* {*hawk/kite*}, which is 'a {hawk/kite} which has a tail like a swallow's' (Allen 1978, pp. 290–1; see also Adams 2001, p. 95 for analogous instances and Norrick 2010 for discussions on this type's scalarity and metaphoricity). This proprietive–similative (polysemy) relationship is a significant semantic factor and will be discussed in detail in Section 5.3.

*3.4. Morphosyntactic Properties*

In contrast to the considerable attention devoted to the phonological and semantic properties discussed earlier, the morphosyntactic properties of *-ed* Adjs have generally received scant exploration in previous studies. Nevertheless, scholars such as Allen (1978), Gram-Andersen (1992), Koenig and Launer (1997), Adams (2001), and Nevins and Myler (2014) have meticulously examined these properties and delineated their idiosyncratic morphosyntactic characteristics.

Initially, it is recognised that the base of the second element in *-ed* compound Adjs can be either a noun or verb, as illustrated in (18) (examples from Adams 2001, pp. 95–96; see Allen 1978, §4.3.3.4 for more data and discussions).

| | | |
|---|---|---|
| (18) | a. | Are the British *colour-prejudiced*? |
| | | 'imbued with colour prejudice' / 'prejudiced with regard to colour' |
| | b. | his *oak-panelled* study |
| | | 'having oak panels' / 'panelled in oak' |
| | c. | the *pedal-powered* drive chain |
| | | 'having pedal power' / 'powered by pedals' |

This issue has engendered multiple analyses and extensive discussions in Levin and Rappaport Hovav (1986), Borer (1990), Dubinsky and Simango (1996), and Bruening (2014). Addressing this categorial ambiguity, Hirtle (1969, p. 25) suggests that "the two morphemes remain distinct, one (that of the past participle) belonging to the grammatical morphology of English, the other (that of the adjective) belonging to its lexical morphology". Gram-Andersen (1992, p. 77) briefly notes that the perfective aspect of the participles of verbs can denote a condition or state (i.e., resultative) resulting from a completed action and that, in the latter, the participle is identical to the *-ed* Adj not only in form but also in meaning. However, this matter is not further pursued here as it lies beyond the scope of this study.

The remainder of this section proceeds in light of the criteria for RAdjs, summarised in (3) in Section 2. The criteria are reiterated in (19) for facile reference.

(19)    a.    Predicability:    *\*this output is industrial, \*this decision is senatorial*
        b.    Gradability:    *\*a very industrial output*
        c.    Comparability:    *\*more industrial*
        d.    Coordinability:    *\*the big and wooden table*
        e.    Quantifiability:    *monochromatic, binational, triconsonantal, multiracial*
        f.    Nominalisability:    *??presidentialness, ??racialness*

First, as confirmed by (19a), RAdjs do not exhibit the predication possibility. However, this characteristic cannot be simply confirmed in *-ed* Adjs, as demonstrated in (20) from Nevins and Myler (2014, p. 248) and (21) from Tsujioka (2002, p. 163).

(20)    a.    John is bearded.
        b.    John is moneyed.

(21)    a.    John is blue-eyed.
        b.    Mary is dark-haired.

Clearly, both *-ed* Adjs derived from bare nouns in (20) and from complex nouns in (21) can appear in a predicate position (see also Beard 1981b, p. 120). Nevins and Myler (2014) contend that *-ed* Adjs can pass the *seem* test (e.g., *John seems blue-eyed* (*in this light*)), establishing their truly adjectival nature. In this regard, Gram-Andersen (1992, p. 164; cf. ibid., pp. 31–32) provides several examples from diverse literary texts, as illustrated in (22).

(22)    a.    In philosophy, where truth seems *double-faced*, there is no man more paradoxical than myself.
        b.    The College became definitely *ill-tempered*.
        c.    For her sake I strove to appear *light-hearted*.
        d.    You shall have a present and not remain *empty-handed*.
        e.    It seemed simple enough. In fact, it seemed downright *simple-minded*.

However, compared to these observations, there are instances where the predicability cannot be confirmed. Compare the examples in (23), cited from Nevins and Myler (2014, p. 248).

(23)    a.    John is moneyed.
        b.    \*John is carred.

Evidently, (23a) is acceptable, whereas (23b) is not, despite the fact that both base nouns do not have an inalienable nature. Furthermore, the complex *-ed* Adjs in (24) also cannot appear in a predicate position (Tsujioka 2002, p. 163).

(24)    a.    \*John is white-housed.    (intended: John has a white house.)
        b.    \*John is big-carred.    (intended: John has a big car.)

As an illustration, from the perspective of informativity, although *carred* in (23b) is rendered sufficiently informative through adding another adjective (*big*, as in (24b)), it remains unacceptable. In relation to predicability, the concepts of inalienability and informativity, which have been extensively debated in prior research, do not seem to provide a satisfactory resolution to this issue.[15]

    Second, whereas RAdjs do not show gradability and comparability, as observed in (19b) and (19c), this does not seem to be the case for *-ed* Adjs. As evidenced in (25), complex *-ed* Adjs are amenable to modification by the degree adverb *very*, indicating their qualitative (or open-scale) status.

(25)    a.    John is very foul-mouthed.[16]    (Nevins and Myler 2014, p. 243)
        b.    She is very kind-hearted.    (Gram-Andersen 1992, p. 37)
        c.    You . . . will end up very narrow-minded.    (Gram-Andersen 1992, p. 164)

Moreover, *-ed* Adjs can manifest comparative forms in (26) and superlative forms in (27), respectively (examples cited from Gram-Andersen 1992, pp. 22–29).

(26) Comparative:
(inflectional) larger-hearted, smaller-boned, harder-headed, shorter-winded, sweeter-tempered, greyer-skinned, truer-hearted, brighter-eyed; (periphrastic) more spirited than ever, more vulgar-minded, more old-fashioned, more narrow-minded, more good-humoured, less ready-witted

(27) Superlative:
(inflectional) largest-hearted, truest-mannered, sharpest-sighted, swiftest-footed, longest-haired, mightiest-brained, coolest-headed, dullest-witted; (periphrastic) the most spirited person, most vulgar-minded, most old-fashioned, most narrow-minded, most renowned, the best-natured

However, as noted in (16) from the prior section, *-ed* Adjs can be considered a close-scale type, thus typically resisting degree modification. Similarly, concerning the comparative degree, *bearded*, for instance, lacks gradability and thus resists modification by the comparative modifier *more*, as shown in (28), taken from Koenig and Launer (1997, p. 66).

(28) ??Santa Claus is more bearded than Lenin.

Third, as regarding the coordination possibility, whereas RAdjs cannot be coordinated with QAdjs, as in (19d), the *-ed* Adjs in (29) are conjoined with the RAdjs *anarchic* and *aquiline* (the suppletive adjectival form of *eagle*), and they do so in (30) with the QAdjs *tall* and *large*.

(29) a. the culture is *anarchic* and *free-spirited*          (Adams 2001, p. 94, italics mine)
     b. Ramrod-tall, *blue-eyed* and *aquiline*, with a high forehead . . .

(30) a. When the *tall* and *bearded* careers advisor. . .
     b. . . . was at the centre of his own *large* and *warm-spirited* family
                    ((29b), (30): Wordbanks Online n.d., italics mine)

Generally, different types of adjectives cannot be coordinated, such as *the fierce and yellow dog* and *the red and sturdy house* (Koenig and Launer 1997, p. 68). Still, the examples in (29) and (30) lead us to surmise that *-ed* Adjs have flexible coordinability.

Fourth, in terms of quantifiability in (19e), *-ed* Adjs exhibit the same behaviour as RAdjs. Koenig and Launer (1997) explain that the proprietive meaning of *bearded*, 'having a beard', is "similar to *dead* and *pregnant*, one either is bearded or one is not" (p. 66). This remark can align with the three-way distinction: (i) BINARY OPPOSITIONS (e.g., *dead*/*alive*); (ii) MULTIPLE OPPOSITIONS (e.g., primary colour terms); and (iii) POLAR OPPOSITIONS (e.g., *rich*/*poor*) (Leech 1974, p. 106ff; cf. Levi 1978, p. 20). In this regard, *-ed* Adjs can be classified under the second category (i.e., multiple oppositions), thus permitting their combination with quantificational (or numeral) prefixes such as *mono-*, *bi-*, *multi-*, and so forth, as enumerated in (31). The data are cited from OED (2023) and Webster's (1913).

(31) a. *mono-*:         monoeyed, monolegged, monocoloured, monoped, monocled
     b. *uni-*:           unilobed, unisexed, unicelled, unifaced
     c. *bi-*:            biforked, biparted, bisexed, bi-lingued, bilobed, bivaulted
     d. *tri-*:           tri-pointed, tri-shaped, tri-membered, trifasciated, tri-sceptred
     e. *quad-*/ *quarter-*:   quadrisulcated, quadrigabled, quatrefoiled, quarter-pierced
     f. *penta-*:         pentahydrated
     g. *hexa-*:          hexaped
     h. *septi-*:         septi-fronted, septi-coloured
     i. *multi-*:         multieyed, multiroomed, multicored, multi-engined
     j. *poly-*:          polychromed, polyspored, polyped, polytyped, polybuttoned
     k. *demi-*/*semi-*:  demi-piqued, demi-natured, semi-calcined, semi-fasciated
     l. *omni-*:          omnitooled

This property markedly distinguishes *-ed* Adjs from QAdjs, as most of the latter fall into binary or polar oppositions, precluding occurrence with numerical prefixes such as *mono-high*, *bi-red*, *tri-strong*, *quadralow*, *multidense*, *polynear*, and *omnistupid* (Levi 1978, p. 24).[17]

Finally, serving as genuine derivatives, *-ed* Adjs can undergo further nominalisation with the *-ness* suffix, as illustrated in (32), which appears to be a diametrically opposite characteristic to RAdjs in (19f).

| | | |
|---|---|---|
| (32) | a. | beardedness, nakedness, wickedness, crookedness, craggedness, forkedness, ruggedness, honeyedness, multicolouredness (OED 2023, s.v. *-ness suffix*) |
| | b. | tight-fistedness, level-headedness, short-sightedness, wrong-headedness (Allen 1978, p. 240); cold-bloodedness, kind-heartedness, left-handedness (Gram-Andersen 1992, p. 38) |

Consequently, *-ed* Adjs potentiate further nominal affixation, providing decisive evidence for the derivative function of the *-ed* suffix.

### 3.5. Summary and Problems of -ed Adjectives

The characteristics of *-ed* Adjs that have been examined thus far suggest that they appear to have an intermediate or dual-standard adjectival status situated between RAdjs and QAdjs. This is evident when comparing their properties to the canonical features of RAdjs presented in (19). This hybrid status can be encapsulated, as in (33).

| | | | |
|---|---|---|---|
| (33) | a. | Predicability: | *the man is bearded* vs. *\*the man is white-housed* |
| | b. | Gradability: | *very foul-mouthed* vs. *\*very handed* |
| | c. | Comparability: | *more spirited* vs. *??more bearded* |
| | d. | Coordinability: | *tall and bearded* vs. *anarchic and free-spirited* |
| | e. | Quantifiability: | *monolegged, bicorned, tri-pointed, multiroomed* |
| | f. | Nominalisability: | *beardedness, nakedness, kind-heartedness, left-handedness* |

With respect to the first three properties outlined in (33a–c), some *-ed* Adjs exhibit these traits while others do not. In (33d), *-ed* Adjs can be coordinated with QAdjs and RAdjs. Additionally, while *-ed* Adjs demonstrate parallel behaviour with respect to quantifiability, as indicated in (19e) and (33e), they differ distinctly in terms of nominalisability when comparing (19f) and (33f).

Consequently, at this point, it appears plausible to suggest that *-ed* Adjs occupy an intermediate status between RAdjs and QAdjs, displaying a subtle departure from the quintessential properties of generic RAdjs. In other words, the classification of *-ed* Adjs under specific adjectival categories remains inconclusive due to their intermediate status. Moreover, previous studies have failed to reach a consensus on whether *-ed* Adjs align more closely with RAdjs or QAdjs, thereby providing ample opportunity to deepen our understanding of adjectivalisation with the *-ed* suffix. Although the ambiguous adjectival status of *-ed* Adjs initially obfuscates our comprehension, a thorough examination of their distinctive characteristics as QAdjs can shed light on this issue.

## 4. The Adjectival Status of *-ed* Adjectives and Their Suffixal Properties

Expanding upon the findings presented in the preceding section, in this section, I further investigate the adjectival status of *-ed* Adjs and provide a clear classification for these adjectives. I contend that *-ed* Adjs should be genuinely classified as RAdjs, adding that their QAdj-like behaviours become apparent through an analysis of conversion, which is discussed in detail in Section 5.

### 4.1. -ed Adjectives Are Intrinsically Relational Adjectives

Upon examining the observations in Section 3, it can be provisionally concluded that *-ed* Adjs bear a closer resemblance to RAdjs than to QAdjs. This is because some *-ed* Adjs have not exhibited QAdjs-like properties, such as predicability, gradability, and comparability, as summarised in (33a–c). These three properties are, in fact, considered the most distinct and prototypical characteristics of RAdjs, a view that is widely shared amongst the scholars mentioned in the Introduction. Furthermore, in terms of quantifiability, as indicated in (33e), *-ed* Adjs permit a range of numerical prefixes in a manner similar to nouns, as illustrated by examples such as *monoplane, biped, triangle, quadrangle, multicylinder*,

and *polysyllable* (Levi 1978, p. 24). My contention is simply based on the fact that, because all the properties of RAdjs (as outlined in (19)) are analogous to those of nouns, if the adjectives in question display nominal properties in certain respects, it is appropriate to classify them as a type of RAdj. However, a rationale for this is necessary.

Nagano (2015, §6), from the perspective of the *Weak Lexicalist Hypothesis* (which deals with inflection in syntax), explains the interesting asymmetry between RAdjs and QAdjs. Nagano argues that, while derivatives with RAdj-forming suffixes (e.g., *-al*, *-ary*, *-ic/-ical*) can additionally possess QAdj properties, such as *grammatical* of *a grammatical sentence* in (2a) (cf. Nikolaeva and Spencer 2013, p. 224; Shimamura 2014, §4.3.5), those with QAdj-forming suffixes (e.g., *-ful*, *-ish*) never concurrently possess RAdj properties. This asymmetry arises because properties that are inherited from the base through derivation *can* have the potential to be lost over time, whereas properties that were not inherited through derivation *cannot* subsequently emerge (Nagano 2015, p. 17). Nagano then underscores that her analysis can be extended to deverbal nominalisations in a similar manner.

The degree of property inheritance from the base noun between RAdjs and QAdjs draws a parallel between the distinction observed within two categories of deverbal nouns, specifically event nominals (hereafter ENs) and result nominals (hereafter RNs). ENs exhibit significantly more verbal characteristics of their base verbs compared to RNs, and, while RNs can refer to a concrete entity, ENs refer to an event (Grimshaw 1990, p. 49). We can compare the pairs of deverbal nominals in (34) and (35), as elucidated by Grimshaw (1990, pp. 49–50, italics mine).

(34) a. The *examination* of the patients took a long time.
   b. The *examination* was on the table.

(35) a. The constant *assignment* of unsolvable problems is to be avoided.
   b. The *assignment* is to be avoided.

For example, *examination* in (34a) serves as an EN, signifying an event and taking the argument *patients* as its underlying transitive verb *examine*. In contrast, *examination* in (34b) functions as an RN, possessing a referential meaning while lacking an argument structure. This distinction is similarly exemplified in the usage of *assignment* in (35a) and (35b), with the former serving as an EN and the latter as an RN. Hence, the two types of deverbal nominals differ in the extent to which they inherit verbal properties from their bases.

Moreover, verbal gerunds (e.g., *destroying* in (36) cited from Spencer and Nikolaeva 2017, p. 80), compared to ENs (e.g., *assignment* in (35a)), express more verbal properties. For example, they take obligatory arguments (see Grimshaw 1990, pp. 50, 56 for other properties).

(36) The Government's/Their systematically
   *destroying* all the evidence appalled us.

Similar to a verb, *destroying* in (36) undergoes modification by an adverb (i.e., *systematically*) and directly takes a direct object (i.e., *all the evidence*). These verbal properties are only observed in gerunds, not in complex event nominals. In contrast, *assignment* in (35a) is fully nominalised, has a referential reading, is modifiable by adjectives (i.e., *constant*) instead of adverbs, and is incapable of directly taking a grammatical object of its base verb.

Interestingly, however, even *-ing* nominals have referential readings. Whereas Grimshaw (1990, p. 56) regards *handwriting* as a lexicalised case and does not discuss the referential reading of *-ing* nominals in detail, Andreou and Lieber (2020, §5) and Lieber and Plag (2022, pp. 327–28) argue that referential readings of *-ing* nominals are viable, though there exists an evident proclivity for numerous *-ing* nominals to denote eventive readings. Lieber and Plag (2022) propose that the referential readings of *-ing* nominals, such as *rock-climbing*, *medical assisting*, and *bungee jumping*, are expressed when they denote "[t]he name for a kind of hobby, pastime, occupation, or job" (p. 315). In addition, the *-ing* nominals *covering(s)* in (37) and *cutting(s)* in (38), taken from Andreou and Lieber (2020, p. 347, bold mine), clearly exhibit referential readings.

(37)  covering / coverings

    a.    *Military History 2005*: Finally he would add a waterproof **covering** of thin leather, bark or snakeskin.

    b.    *Saturday Evening Post 1993*: You want bulbs without deep scars, but small nicks and loose paperlike **coverings** (tunics) are acceptable.

(38)  cutting / cuttings

    a.    *Horticulture 1992*: Firm the moss around the **cutting** to protect it until it roots.

    b.    *Harpers Magazine 1993*: His wife walked in from outside, carrying some **cuttings**.

    c.    *Ebony 2000*: 'Ye shall not make any **cuttings** in your flesh for the dead, nor print any marks upon you'.

For instance, the singular form *covering* in (37a) and the plural form *coverings* in (37b) denote the same meaning, 'a layer that covers something'. Both *cutting* in (37a) and *cuttings* in (38b) similarly express 'something that has been cut off (e.g., a part of a plant)'. However, *cuttings* in (38c) means 'an injury made when the skin is cut with something sharp'. Andreou and Lieber (2020, p. 347) explicate that this meaning is similar to the converted form, *cuts*, and not derived from the singular form, *cutting*. Furthermore, they observe that some plural *-ing* nominals such as *winnings*, *takings*, *meltings*, and *drippings* have quite distinct meanings (e.g., a collective reading) from their singular counterparts (ibid., pp. 347–48). The polysemy/polyfunctionality can therefore be observed not only in Latinate nominalising suffixes such as *-tion* and *-ment*, but also in the Germanic *-ing* suffix.[18]

    Accordingly, it appears that nominalisations derived as ENs can extensionally acquire RN usage, thus losing the properties of their base verbs. Conversely, it seems rare for nominalisations derived as RNs to extend to the usage of ENs; namely, it is uncommon for them to acquire verbal characteristics at a later stage.[19] Turning back to *-ed* Adjs, it is reasonable to presume that *-ed* adjectivalisations as RAdjs can subsequently acquire the usage of QAdjs, but the contrary is not feasible. Hence, it is appropriate to regard *-ed* Adjs as an intrinsic type of RAdj.

### 4.2. -ed *as a 'Dedicated' Relational Adjective Suffix*

    Given that *-ed* Adjs are unequivocally RAdjs, the following question arises: Is there any divergence between general RAdjs and *-ed* Adjs? I argue that considering the etymology and semantic properties of the suffixes can be a critical factor and offers a plausible answer to this question. Specifically, along with Rainer's (2013) two distinct types of suffixes, while Latinate suffixes are considered *all-purpose* relational suffixes, native suffixes are regarded as *dedicated* relational suffixes. This analysis leads to the corollary that *-ed* is a dedicated RAdj suffix.[20]

    The different properties between the two types of deverbal nominals (see Section 4.1) can be attributed to the features of their suffixal etymology. For instance, *assignment* in (35b) exhibits true nominal behaviour as an RN, whereas *destroying* in (36) shows more verbal behaviour as a verbal gerund. Paying careful attention to the suffixal etymology of these examples, it is important to acknowledge that *-ment* possesses the [+Latinate] feature and *-ing* has the [−Latinate] feature. Based on this and Nagano's (2015) analysis, Ishida et al. (2021) (see also Ishida 2021) hypothesise that the etymological difference between suffixes (whether Latinate or Germanic) can be identified in the lexicon with the [±Latinate] feature (cf. Giegerich 1999). For example, RAdjs with [+Latinate] suffixes and [−Latinate] suffixes can be depicted as in (39).

(39)     a.     RAdjs with [+Latinate] suffixes:     *senatorial, polar, algebraic*

        b.     RAdjs with [−Latinate] suffixes:     *bearded, wooden, southern*

Thus far, the [±Latinate] feature of suffixes seems to distinguish between the two types of RAdjs (see Section 4.3 for discussion of native suffixes other than *-ed*), but it remains unclear how this is related to the divergence between general RAdjs and *-ed* Adjs. To clarify this, another perspective on the suffixal property should be examined.

    Rainer (2013, §1.4) divides denominal adjectivalising suffixes into the two categories of *all-purpose* and *dedicated*. All-purpose suffixes are essentially relational and encompass

various kinds of semantic relations. In contrast, dedicated suffixes express "only one specific relation or a small set of such relations" (Rainer 2013, p. 16), and they are therefore, in theory, subsumed under the all-purpose suffix(es). Based on this suffixal dichotomy, Latinate suffixes that form RAdjs can be categorised as all-purpose suffixes, as RAdjs with these suffixes exhibit the same range of denotations as their base nouns. Recall that linguists do not typically assign any meaning to the suffixes of RAdjs or, at most, attribute a highly abstract meaning to them; namely, the absence of semantic predicates in Latinate suffixes results in a contextually determined meaning of RAdj-N combination, being akin to compounds (see (1)). Conversely, the suffix *-ed* can be classified as a dedicated one, as it primarily expresses a *proprietive* meaning. Thus far, in deriving RAdjs, two types of suffixal properties and their functions can be identified, as shown in (40).

(40)  a.  If the suffix of an RAdj has the [+Latinate] feature, it serves as an all-purpose suffix and expresses genuinely relational semantics.

b.  If the suffix of an RAdj has the [−Latinate] feature, it serves as a dedicated suffix and expresses a certain specific meaning (e.g., proprietive).

Accordingly, the [±Latinate] feature specification underscores the disparity between general RAdjs and *-ed* Adjs in ascertaining the type of RAdj that can be derived from a given suffix.

One may wonder how both proprietive and relational properties coexist in *-ed* Adjs, or which property of the two is their primary characteristic. The concurrent properties should be categorised into two different aspects, i.e., a purely semantic property and a modifying function. The former directly corresponds to the proprietive meaning of *-ed* Adjs and the latter occurs collaterally through their classificatory function in noun modification. The rationale for the dual semantic relationship can be associated with meronymic (or part-whole) relations between the base nouns of *-ed* Adjs and the head nouns (cf. Nikolaeva and Spencer 2020, p. 102; see also Nagano 2023, §3.3). It is important to note that meronymy is a distinct concept from hyponymy, as hyponymy expresses a *kind* relation, whereas meronymy expresses a *part* relation; thus, "a dog is a kind of animal, but not a part of an animal; a finger is a part of a hand, but not a kind of hand" (Cruse 2006, pp. 105–6). A meronymic relation is closely related to the input categories of *-ed* Adjs, given that they are typically derived from relational nouns such as kinship (e.g., *brother, husband*), social non-kinship (e.g., *friend, governor*), operational (e.g., *picture, rumour*), relative part (e.g., *edge, corner*), body part (e.g., *leg, eye*), and properties (e.g., *height, colour*) (cf. Newell and Cheung 2018). Relational nouns are known to pertain to a particular entity based on its association with another entity (Barker 2011).

Scholars, such as Takehisa (2017), Ishida (2023), and Morita (2024), argue that the input for *-ed* Adjs is restricted to relational nouns.[21] This strongly suggests that scholars should connect relational nouns with the notion of the (in)alienable possession of *-ed* Adjs (see Introduction), as indicated by Spencer (2018), and the input property of *-ed* Adjs is generally regarded as inalienable rather than alienable. On the basis of these considerations, the particularly prominent semantic property of *-ed* Adjs (i.e., proprietive) prevails in their adjectival status as a dedicated RAdj suffix over the relational meaning. This outcome is consistent with the etymological feature of the suffix (i.e., Germanic) and aligns with Rainer's (2013) suffixal dichotomy.

### 4.3. Relational Adjectives with Other Germanic Suffixes

The above conclusions pique our interest in similar native RAdjs in English, such as those formed by the suffixes *-en* (e.g., *wooden*) and *-ern* (e.g., *southern*). I argue that these Germanic suffixes should also be regarded as dedicated RAdj suffixes, similar to *-ed* (see (39b)).

Regarding the suffix *-en*, Adjs derived from this suffix (hereafter *-en* Adjs) express specific semantics, such as 'made of X, consisting of X', where X represents material-denoting nouns (e.g., *wooden, woollen, earthen, wheaten*) (Marchand 1969, pp. 270–71; see Ishida et al. 2021 for more detailed discussions about *-en* Adjs). Similar to *-ed* Adjs, the

material semantics of *-en* Adjs clearly suggests that the suffix is a dedicated one, in contrast to Latinate suffixes (see (39a)).

As for another Germanic suffix, *-ern*, Marchand (1969, p. 271) explains that this only forms adjectives from nouns denoting points of the sky, such as *northern*, *eastern*, *southern*, and *western* (hereafter *-ern* Adjs), and it appears to be a historic extension of the suffix *-en*. Although a definitive conclusion regarding *-ern* as another type of dedicated suffix requires further exploration, Marchand's account suggests that it possibly represents a specific meaning, such as 'situated in X, towards X' (see OED's (2023) account of those *-ern* Adjs).

Therefore, including *-ed*, the discussion about dedicated suffixes thus far can be summarised in (41).

(41)   a.   *-ed*:      proprietive ('having X, being provided with X')
       b.   *-en*:      material ('made of X, consisting of X')
       c.   *-ern*:    locational ('situated in X') / directional ('towards X')

In comparison to these, comparison suffixes such as *-al*, *-ary*, or *-ic/-ical* are purely relational in nature since they do not exhibit specific semantics. This proposition bolsters my contention in (40) and offers a new perspective on RAdjs in the English language.

One might question whether *-less* can be considered the opposite counterpart to *-ed*; however, this is not the case. The true counterpart of *-less* is the suffix *-ful*, as denominal adjectives derived from *-less* and *-ful* can be inherently QAdjs based on their QAdj-properties. In this regard, the suffixes *-less* and *-ful* are dedicated QAdj suffixes. As discussed by Rainer (2013, p. 17), RAdjs seemingly never convey the meaning 'without N', which seems likely to be why numerous languages possess dedicated privative suffixes, such as English *-less*, German *-los*, and so forth (see Footnotes 8 and 9 for relevant discussions).

*4.4. Summary and Additional Remarks*

Hitherto, an examination of the adjectival status of *-ed* Adjs and their suffixal properties has yielded the following succinct synthesis:

- *-ed* Adjs are intrinsically RAdjs, as evidenced by their property inheritance from base nominals.
- *-ed* Adjs distinguish themselves from other RAdjs by virtue of their native (i.e., Germanic) suffixal etymology, which contrasts with the non-native (i.e., Latinate) origins of their RAdj counterparts.
- *-ed* is a dedicated RAdj suffix (i.e., proprietive) together with other Germanic suffixes, such as *-en* (i.e., material) and *-ern* (i.e., locational/directional).

It is pertinent to address the question of why only RAdjs formed by the native suffixes are considered to be meaning-bearing RAdjs while those formed by non-native (i.e., Latinate) suffixes are not. The answer may reside in the inherent multifunctional nature of native suffixes, a characteristic that non-native suffixes generally lack due to their limited ability to penetrate the inflectional domain. For example, the suffix *-ed* exhibits versatility as a regular past-tense marker, a past-participle marker, and a denominal adjectivaliser. In addition, *-en/en-*, which also serves multiple functions, appears as a verbal prefix (e.g., *enlighten*), diminutive marker (e.g., *chicken*), feminine marker (e.g., *vixen*), plural marker (e.g., *oxen*), deadjectival verbaliser (e.g., *darken*), participle marker (e.g., *broken*), and denominal adjectivaliser (e.g., *wooden*). Originally an inflectional element in Old English, *-en* has, probably through 'being copied', established its multifunctional status, much like *-ed*. As for *-ern*, it is no longer productive and does not exhibit any multifunctional nature at all (cf. Marchand 1969, p. 226; Bauer et al. 2013, p. 304). Conversely, it is challenging for Latinate suffixes forming RAdjs to acquire such multifunctionality, owing to their borrowed nature (see Ishida 2021, §5.4.2.2 for an in-depth discussion on this matter). Therefore, an examination of suffixal multifunctionality requires meticulous scrutiny and should be systematically investigated in detail.

As an additional remark, although it will not be discussed in detail, another Germanic suffix, *-ly*, should be mentioned. This suffix derives adjectives not only from base nouns of persons that denote 'in the manner of X, like an X', as in *brotherly*, *daughterly*, *fatherly*, and *womanly*, but also from temporal nouns (e.g., *half-hourly*, *daily*, *monthly*) and directions (e.g., *easterly*, *southwesterly*) (Plag 2018, p. 97; Marchand 1969, pp. 329–31; cf. Bauer et al. 2013, pp. 304–6). Crucially, Marchand (1969, §4.64.3) observes that, in Old and Middle English, the suffix denotes a neutral (i.e., relational) sense 'characteristic of X, belonging to X', as in *deathly*, *fleshly*, *heavenly*, *lively*, *lovely*, *summerly*, *homely*, and *shapely*; namely, the semantics are still the same as those of RAdjs. This fact seems to support my contention regarding native suffixes (i.e., *-ly* is another type of dedicated RAdj suffix). However, a more detailed analysis is necessary, such as an examination of the differences and competition between the forms of *-ly* Adjs and suppletive RAdjs (e.g., *fatherly* vs. *paternal*; *summerly* vs. *aestival*; *nightly* vs. *nocturnal*).

## 5. The Qualitative Adjective(-like) Behaviour of *-ed* Relational Adjectives

The puzzle of QAdj(-like) behaviour of *-ed* Adjs still persists. As previously mentioned in (2) of Section 2, we have observed two types of *grammatical*. Recall that *grammatical* in (2b) serves as an RAdj, while, in (2a), it functions as a QAdj. The question arises as to how these two distinct adjectival categories can be reconciled. This section employs a conversion analysis to provide an answer to this question and explores the very relationship between *-ed* Adjs as RAdjs and those as QAdjs.

### *5.1. Nagano's (2018a) Conversion Analysis*

Bauer and Valera (2005) discuss the definition of conversion as a word-class change without a form change (e.g., *a bridge* and *to bridge*), emphasising the importance of considering "how narrowly a word-class is to be defined" (p. 10) (cf. Valera 2014, 2021; Martsa 2021). They question whether homophonous pairs *within* a word-class (often described as *secondary word-class conversion*) can be considered conversion cases, such as those in (42) (cf. Bauer et al. 2013, pp. 557–59).

|       |    |            |                          |                                     |
|-------|----|------------|--------------------------|-------------------------------------|
| (42)  | a. | Noun:      | proper vs. common        | e.g., *Mary* vs. *the four Marys*   |
|       |    |            | countable vs. uncountable | e.g., *a cake* vs. *some cake*     |
|       |    |            | concrete vs. abstract    | e.g., *a beauty* vs. *beauty*       |
|       | b. | Verb:      | intransitive vs. transitive | e.g., *to walk* vs. *to walk a dog* |
|       | c. | Adjective: | relational vs. qualitative | e.g., *English* vs. *very English* |

Although investigating the definition of conversion is not the primary purpose here, Bauer et al. (2013, pp. 557–59), for instance, regard these examples as cases of *type-shift* (or *coercion*; cf. Pustejovsky 1995) rather than conversion. For example, Bauer et al. (2013, p. 318) state that most RAdjs can be semantically coerced into a qualitative interpretation; however, their definition of *qualitative reading* remains unclear (Nagano 2018a, pp. 185–86). For example, it is uncertain whether they are implying that RAdjs can be coerced into similative adjectives. Nagano (2018a, p. 187) examines RAdj-to-QAdj alternation as a conversion rather than a coercion. Based on her analysis in the following, I similarly contend that the RAdj-to-QAdj alternation in (42c) constitutes a type of conversion.

Note that the QAdj(-like) behaviour of *-ed* Adjs has been clearly ascertained from Section 3. I then propose a more refined perspective, having argued that *-ed* Adjs are intrinsically RAdjs, with QAdj properties being subsequently acquired via conversion. This assertion gains efficacy when examining *-ed* Adjs from the vantage point of QAdjs rather than exploring them in the realm of RAdjs. The differentiation between homophonous pairs of RAdjs and QAdjs has previously been underscored by Bauer et al. (2013, p. 318), as illustrated in (43).

(43)  a.  *Newsweek 1997*: France is second—75 percent of French electricity **is nuclear**, which has reduced French air pollution fivefold—followed by Russia and Japan.

  b.  *USA Today 2005*: The outspoken Texas conservative, who displays the Ten Commandments in his office but admits he has a hard time loving his enemies, declined to run for House speaker in 1998 because he considered himself "**too nuclear**".

Interpreting these examples, it becomes apparent that *nuclear* in (43b) undeniably carries a qualitative connotation (e.g., 'emitting dangerous radiation') and conveys a property amenable to gradability. In contrast, *nuclear* in (43a) lacks this characteristic. Instead, the latter instance preserves the basic classificatory function inherent in RAdjs. Thus, the sentence in (43a) signifies that 75 percent of electricity generated in France pertains to the nuclear kind. Here, the expression *75 percent* in (43a) is a quantifier (cf. *primarily*, *mostly*, *mainly*, *all*), not a degree head. It specifies the proportion of the subject referent that belongs to the class denoted by the predicative complement NP. This interpretation is supported by the *class-membership* sentence without an adjective, as in *This object is partially a bed* (Fábregas 2014, p. 284). Thus, *nuclear* in (43a) is an RAdj and indicates the class membership of *French electricity*, while that in (43b) is a QAdj and expresses that the subject has a particular property represented by the adjective.

Addressing the issue in (43), Nagano (2018a, p. 196) expounds that the two manifestations of *nuclear* in (43) can be demonstrated through intransitive structures in (44a) and (44b), respectively.

(44)  a.  *nuclear* in (43a):  Subject$_i$ + be + [RAdj + one$_i$ /$\emptyset_i$]$_{NP}$
  b.  *nuclear* in (43b):  Subject + be + [Deg [QAdj]$_{AP}$]$_{DegP}$

Building upon Levi's (1978, §7.2) intricate exposition, Nagano (2018a) argues that RAdjs occupying predicate positions results from the ellipsis originating in their initial prenominal position (cf. Nagano 2016, 2018a; Shimada and Nagano 2018; Ishida 2020, 2021; Ishida and Naya 2022). As delineated in (44a), *nuclear* in (43a) does not operate as a true adjectival predicate akin to QAdjs in (43b); rather, it stands as a stranded prenominal modifier, left by coreferential *one*-substitution (i.e., *75 percent of French electricity is the nuclear one*) or bereft of its corresponding modifying target (i.e., the head noun) due to deletion.

Nagano (2018a) advances a convincing mechanism for the RAdj-to-QAdj conversion process. Her central proposition is that an RAdj within an NP "is raised into a QAdj slot through the force of the modal attributive *true*" (Nagano 2018a, p. 206), as depicted in (45) from Nagano (2018a, p. 202).

(45)  Input structure  Output structure
  [true [RAdj + N]$_{NP}$]$_{NP}$  >  [(truly) QAdj [. . .]$_{NP}$]$_{NP}$

The impact of *true*-modification of RAdjs serves to accentuate the assertion of *classificational truthfulness*. As is widely acknowledged, modal attributes provide a specific value for the truthfulness operator associated with their host NP (cf. Pullum and Huddleston 2002). This assessment determines the accuracy of the classification of an entity or concept. For example, the modal attributive *apparent* in *an apparent electrical fire* "does not modify any stated ascription (it was not apparent fire or apparent electricity), but the implicitly ascribed meaning 'caused by'" (Feist 2012, p. 85). This reveals the fact that the expression denotes "a fire that is apparently *caused by* electricity" (Nagano 2018a, p. 201), where *apparent* also takes an adverbial form, *apparently*, in the related clausal structure (cf. *the actual cause* 'that which is *actually* the cause'; *a certain winner* 'one who will *certainly* be a winner', Pullum and Huddleston 2002, p. 557). On this basis, Nagano (2018a) applies *true* as a modal attributive to the constructed RAdj+N combination. The examples in (46) are taken from Nagano (2018a, p. 202).

(46)   a.   *true* electrical fire
= fire that can truly be called an electrical fire

        b.   a *true* musical clock
= clock that can truly be called musical clock

        c.   *true* Dalmatian wine
= wine that can truly be called Dalmatian wine

        d.   *true* nuclear weapon
= object that can truly be called nuclear weapon

        e.   a *true* English {woman/$\emptyset_{\text{PERSON}}$}
= person who can truly be called English {woman/$\emptyset_{\text{PERSON}}$}

Nagano explicitly contends that the introduction of *true* and its function of changing semantics precisely represent the process of RAdj-to-QAdj conversion.

*5.2. A Conversion Analysis of -*ed *Adjectives*

We are now in a position to apply Nagano's (2018a) insightful analysis to the case of *-ed* Adjs as QAdjs. Let us first observe how bare N-*ed* Adjs occur in gradable structures in (47). All the sentences are sourced from Wordbanks Online (n.d.).)

(47)   a.   They demonstrated that icecaps were smaller and tundra zones more *wooded* than had previously been imagined.

        b.   Rio Ferdinand is more *cultured* and will relish playing at his Manchester United home.

        c.   The ship's insurers argue that they threw the slaves overboard because they were too *diseased* to be sold.

        d.   The resulting emptinesses are too *mannered* and elegant to be real.

        e.   St. Clair is certainly very much *freckled*, although I try to prevent the others from commenting on it . . . for I was freckled once and well do I remember it.

The italicised bare N-*ed* Adjs are all used as QAdjs, as they straightforwardly show gradability. Employing *true*-modification, we can derive the following interpretations for each case, as indicated in (48).

(48)   a.   *true* wooded tundra zones
= tundra zones that can truly be called wooded tundra zones

        b.   *true* cultured Rio Fernand
= Rio Fernand that can truly be called cultured Rio Fernand

        c.   *true* diseased slaves
= slaves that can truly be called diseased slaves

        d.   *true* mannered emptinesses
= emptinesses that can truly be called mannered emptinesses

        e.   *true* freckled St. Clair
= St. Clair that can truly be called freckled St. Clair

As observed in each pair of interpretations, the internal adjective acquires distinct scalar semantics as a result of the accentuation of truthfulness. Furthermore, the analysis can be applied to complex N-*ed* Adjs, as illustrated in (49).

(49)   a.   a *true* foul-mouthed John               (cf. (25a))
= a man named John who can truly be called foul-mouthed

        b.   a *true* kind-hearted lady               (cf. (25b))
= a lady that can truly be called a kind-hearted lady

        c.   a *true* narrow-minded person           (cf. (25c))
= a person that can truly be called a narrow-minded person

In the previous section, there were only a few points to corroborate our view of regarding *-ed* Adjs as a type of RAdjs; however, upon reflection, the analysis of *-ed* Adjs as QAdjs sufficiently convinces us that *-ed* Adjs are originally RAdjs, as well as that their QAdj behaviour, such as predicability and gradability, can be acquired through RAdj-to-QAdj conversion, which is implemented by *true*-modification (Nagano 2018a). In this way, the idiosyncratic properties of *-ed* Adjs are no longer perplexing.

*5.3. The Semantics of Converted* -ed *Adjectives*

Thus far, *-ed* Adjs exhibiting QAdj behaviour can plausibly be considered the outcome of the morphological process, i.e., RAdj-to-QAdj conversion. This section discusses how *-ed* Adjs can undergo the same conversion process to QAdjs as general Latinate RAdjs (e.g., *grammatical*), exploring the underlying motivation behind RAdj-to-QAdj conversion. Specifically, I argue that there are two semantic factors driving the RAdj-to-QAdj conversion of *-ed* Adjs: (i) class-membership relation and (ii) proprietive–similative polysemy.

In terms of class-membership relations, as reviewed in Section 5.1, RAdjs as classifying modifiers establish a class-membership relation with their head nouns. Nagano (2018a) explains that converted QAdjs deviate from similative (e.g., *-esque*, *-ish*, *-like*) and *-ful* adjectives, as the latter QAdjs are "highly compositional" (p. 197) in nature, being predicated on perceptual similarity (e.g., similative adjectives: [(subject) be close to base noun]; *-ful* adjectives [(subject) be full of base noun]). In contrast, the former converted QAdjs do not exhibit such compositional characteristics (e.g., *nuclear* in (43b) ≠ [(subject) be close to nucleus], ≠ [(subject) be close to nuclear (state)], or ≠ [(subject) be full of nucleus]). Instead, they are grounded in the notion that an entity or individual constitutes a true member of the base class, either as a representative member or a metaphorical class membership. The former case can be *a highly professional professional writer* (Farsi 1968, p. 56; cf. Nagano 2018a, pp. 186, 204), where *professional*, positioned far from the head noun, is a converted QAdj. This usage emphasises that the subject *writer* is a quintessential representative of the class of *professional writers*, showcasing their exceptional skills and expertise in the field. As for the latter, *too nuclear* in (43b) suggests that the individual in question truly emits hazardous radiation, implying that he possesses a dangerous level of nuclear properties or characteristics. This interpretation is based on the notion that the person is a true representative member of the *nuclear* class, metaphorically speaking. Another typical example is the sentence *He's more English than the English* (Quirk et al. 1985, p. 1565; cf. Nagano 2018a, p. 204), which expresses the idea that the person's characteristics encapsulate the genuine essence of Englishness, extending beyond mere resemblance to English people in the appearance or superficial traits. This interpretation indicates that the individual embodies the true nature of Englishness, making them a metaphorical representative member of the *English* class. These inferences are reasonably obtained from the semantic interpretations of *true* modification.

The case of converted *-ed* Adjs can be compared to RAdjs with Latinate suffixes, in that they establish a class-membership relation with their modified head nouns. For example, *wooded* in (47a) expresses that the *tundra zones* are a representative member of a class characterised by having wooded vegetation. The use of the operator *true* in (48a) emphasises this class membership by suggesting that the subject is a genuine example of this class. Metaphorical class membership can also be conveyed through the use of converted *-ed* Adjs, as demonstrated in (50a) and (50b). In these examples, *wooded* and *bearded* are used metaphorically to describe non-literal class membership.

(50)  a.  The room was *wooded* with dark, heavy furniture that seemed to crowd in on them, making it difficult to breathe.

(From *The Thirteenth Tale* by Diane Setterfield)

   b.  The rocky cliff was *bearded* with a thick growth of moss and lichen, giving it a wild and untamed appearance.

(From *The Light Between Oceans* by M. L. Stedman)

In the example in (50a), *wooded* serves as a metaphorical descriptor for the *room*, which is replete with dark, heavy furniture that resembles trees in a forest. In the case of (50b), *bearded* functions as a metaphorical characterisation of the *rocky cliff*, which is covered with moss and lichen that resemble a beard. Furthermore, the case of complex *-ed* Adjs can be found, as shown in (51), where *foul-mouthed* and *kind-hearted* are used to describe inanimate objects and abstract concepts, respectively.

| (51) | a. | The car's engine was *foul-mouthed*, sputtering and cursing as it struggled to turn over. |
|---|---|---|

<div align="right">(From <em>The Girl Who Kicked the Hornet's Nest</em> by Stieg Larsson)</div>

| | b. | The small town was *kind-hearted*, welcoming strangers with open arms and offering help to those in need. |
|---|---|---|

<div align="right">(From <em>The Help</em> by Kathryn Stockett)</div>

In (51a), *foul-mouthed* metaphorically describes the *car's engine*, serving to depict the unpleasant noises emitted by the machine as being akin to cursing and swearing. Conversely, *kind-hearted* in (51b) figuratively characterises the subject *small town* by its friendly and generous inhabitants. These examples effectively illustrate metaphorical class membership through inferences derived from the semantics of the *true* operator.[22]

Overall, the use of converted *-ed* Adjs, whether simple or complex, can serve to establish class membership or convey metaphorical class membership. Such a class-membership relation thus not only explains the mechanism of RAdj-to-QAdj conversion of Latinate RAdjs but also the very motivation for that of *-ed* Adjs. This analysis further strengthens the analysis of defining *-ed* Adjs as a type of RAdj in Section 4.

With respect to proprietive–similative polysemy, the assertion that the *-ed* suffix should be considered a dedicated suffix is grounded not only in its proprietive semantics per se but also in this semantic property. First, the claim that *-ed* Adjs inherently convey a proprietive meaning is founded on their asymmetrical possessive–proprietive relation (see Section 3.3). Unlike other possessive or ethnic adjectives (e.g., *my*/*your*/*her*/*their* (*house*); *Napoleonic* (*era*), *Darwinian* (*theory*)), *-ed* Adjs are uniquely specialised in proprietive semantics. Next, like *-y* Adjs, and as discussed by Nagano (2023, §3.3), proprietive–similative polysemy is discernible in *-ed* Adjs. Nagano (2023, p. 519) applies Fradin's (2007) analysis of French *-eux* Adjs to *-y* Adjs (e.g., *rusty*) and explains that *rusty* is polysemous with two readings: proprietive (as in *rusty knife*) and similative (as in *rusty hair*). In alignment with the observations in (17), *-ed* Adjs in (52) to (54) plainly exhibit similar behaviour; thus, the (a)-examples convey a proprietive meaning and the (b)-examples convey a similative meaning.

| (52) | a. | bearded lady | 'lady having a beard' |
|---|---|---|---|
| | b. | bearded rock | 'rock covered with tufts or appendages like a beard' |

| (53) | a. | shirt-sleeved man | 'man wearing a shirt without a coat or jacket' |
|---|---|---|---|
| | b. | shirt-sleeved weather | 'weather pleasantly warm enough to live or work in clothing like short-sleeves' |

| (54) | a. | long-tailed tit | 'titmouse having a very long tail' |
|---|---|---|---|
| | b. | long-tailed pair | 'electronic circuit configuration in which the output voltage is proportional to the difference between two input voltages, characterised by having one transistor with a significantly larger current than the other, similar to how data points extend out like a long tail in a graphical representation' |

For example, *bearded* in (52a) can be naturally interpreted as proprietive, as the head noun, *lady*, can have a beard as part of her body. In contrast, regarding *bearded* in (52b), although it still retains the general meaning of *-ed*, 'covered with', the semantic relationship between its base and the head noun (i.e., *rock*) cannot be inferred in a proprietive manner, as a rock as a natural object does not possess a human beard; therefore, this triggers an alternative similative meaning (i.e., 'beard-like'). Similar contrastive observations can be applied to the other examples in (53) and (54).

In contrast to the analysis proposed by Nagano (2018a), which posits that converted QAdjs exhibit semantic distinctions from similative and *-ful* adjectives, the previous observations challenge this assertion. I contend that the crux of this issue resides in the suffixal attributes of RAdjs, specifically in regard to whether these suffixes are *all-purpose* or *dedicated* (see Section 4.2). RAdjs with Latinate suffixes undergo conversion into QAdjs, thereby establishing a (metaphorical) class-membership relation through their suffixal all-purpose

properties. Conversely, *-ed* Adjs undergo a similar conversion into QAdjs; however, due to their suffixal dedicated properties and the intrinsic proprietive semantics, as well as the nature of proprietive–similative polysemy, they convey both similative semantics and a (metaphorical) class-membership relation.

Thus, analogous to the case of general RAdjs, *-ed* Adjs also undergo conversion into QAdjs. Nevertheless, *-ed* Adjs differ from typical RAdjs in that their conversions are propelled by the semantics of the dedicated suffix as well as the inference indicated by *true*-modification.

*5.4. More on the Complex Qualitative Adjective Semantics of -*ed *Adjectives*

Ishida (2023) provides further insights into the semantic properties of *-ed* Adjs as QAdjs. Certain QAdjs that undergo conversion from RAdjs can indeed exhibit a more complex figurative extension of their semantics. For instance, *monochromatic*, as an Radj, expresses 'having one colour', but its converted QAdj imbues it with the sense of 'drab, unvarying' (see also Ishida 2020, pp. 39–40, 42 for cases involving non-numeral prefixed RAdjs converted into QAdjs). Such further complex semantic extension can also be evinced in the case of *-ed* Adjs.

Ishida (2023) argues that a specific class of *-ed* Adj, conventionally treated as lexicalised (or idiomatic) cases (cf. Bauer and Huddleston 2002, p. 1709) due to their unpredictable meanings (e.g., *forked road* does not mean 'road with or having a fork' but rather 'a bifurcated, branching road'), retains its intrinsic proprietive semantics. Typical examples are provided in (55), cited from Ishida (2023, p. 64).

| (55) | a. | forked road | = | fork [*-shaped*/*-formed*] road |
|---|---|---|---|---|
| | b. | dogged persistence | = | dog [*-natured*/*-tempered*] persistence |
| | c. | cupped flower | = | cup [*-shaped*/*-formed*] flower |
| | d. | crooked business | = | crook [*-natured*/*-tempered*/*-dispositioned*] business |
| | e. | ragged coat | = | rag [*-natured*/*-typed*/*-patterned*] coat |

For example, if the phrase *forked road* in (55a) is paraphrased with a certain relational noun, such as *shape* or *form*, the semantically corresponding *-ed* Adjs, *fork-shaped*/*formed road*, can be obtained, as provided on the right side of (55a). This analysis can be substantiated by its corresponding *with*-phrase (namely, *road with a fork shape*/*form*) and further enable the avoidance of unwarranted exclusions of such lexicalised cases, considering even these *-ed* Adjs as preserving their inherent proprietive meanings as original RAdjs. Significantly, the derived meaning in this context can be considered reminiscent of the class of similative *-ed* Adjs in (17) (e.g., *forked road* as a 'road with a shape like a fork').

Additional examples can be comprehended in a manner akin to the case of (55). Gram-Andersen (1992, pp. 70–71) highlights that, in certain complex *-ed* Adjs, even if the *-ed* Adjs as their second components are elliptical, the meaning of the first components is essentially preserved and is the same as that of the whole expression, as illustrated in (56).

| (56) | a. | a deep-*toned* voice | = | a deep voice |
|---|---|---|---|---|
| | b. | an oval-*shaped* face | = | an oval face |
| | c. | a kind-*hearted* old lady | = | a kind old lady |
| | d. | a simple-*minded* fellow | = | a simple fellow |

This observation underscores the true qualitative status of *-ed* Adjs as QAdjs, although there are instances that deviate from this pattern (e.g., *a short-armed person* $\neq$ *a short person*). It seems safe to say that Ishida's (2023) analysis and Gram-Andersen's (1992) findings are interrelated, evincing a reverse of the complex semantic pattern exhibited by *-ed* Adjs, as compared in (55) and (56).

Provided that Ishida's (2023) analysis is well-founded, more intricate QAdj semantics of *-ed* Adjs can also be captured in a similar manner. The examples are illustrated in (57), which are cited from Tsujioka (2002, p. 164; cf. Nevins and Myler 2014, pp. 253–54).

| (57) | a. | blue-blooded | = | noble |
|---|---|---|---|---|
| | b. | white-livered | = | timid |
| | c. | close-fisted | = | stingy |
| | d. | hard-headed | = | stubborn |

For example, *blue-blooded* (*man*) in (57a) metaphorically denotes a nobility. The expression distinguishes the upper class from the working class, in that a nobility's superficial veins through untanned skin appear blue due to not engaging in outdoor labour. This can also instantiate the similative case of *-ed* Adjs, namely, *blue-blooded man* being a 'man with blood which looks like blue in colour'. In this manner, Ishida (2023) reveals the very mechanism of semantic extension of *-ed* Adjs, and the discussion of proprietive–similative polysemy further lends plausible support to this analysis. Additional instances are presented below (see also Gram-Andersen 1992, pp. 91–93 for similar examples). The bare N-*ed* in (58) and complex N-*ed* in (59) are taken from OED (2023).

| (58) | a. | cankered | = | malignant, spiteful, envious (cf. *bad-tempered*) |
|---|---|---|---|---|
| | b. | floored | = | overpowered, done for |
| | c. | haltered | = | fettered, hampered |
| | d. | laureled | = | honoured, illustrious |
| | e. | misted | = | dulled, blurred |

| (59) | a. | blue-eyed | = | innocent, ingenuous |
|---|---|---|---|---|
| | b. | quick-eyed | = | perceptive (cf. *sharp-eyed*, *keen-sighted*) |
| | c. | ill-tempered | = | morose, cross, peevish |
| | d. | half-assed | = | contemptible, ridiculous |
| | e. | free-spirited | = | lively and independent; unconstrained by convention |

Although each example is not examined here, as expected, these examples now serve to qualify the referents (predominantly human properties) as true QAdjs. Moreover, for more lexicalised *-ed* Adjs derived from *missing inputs*, such as *wicked*, *naked*, and *jagged*, this semantic extension is considered fixed (or fossilised) over time, alongside the demise of the base nouns (cf. Harley 2006).

*5.5. Summary and Further Remarks*

It is now evident that *-ed* Adjs are essentially RAdjs converted into QAdjs, providing a comprehensive understanding of their inconsistent behaviour. The preceding analysis can be summarised as follows: (i) The conversion process of *-ed* Adjs, in conjunction with Latinate RAdjs, is elucidated by postulating the *true* modal attribute. (ii) Through conversion, both *-ed* Adjs and Latinate RAdjs establish a class-membership relation (either as a representative member or a metaphorical class membership) with their base classes, and (iii) the resultant semantic distinction between Latinate RAdjs and *-ed* Adjs arises from their suffixal semantics (i.e., *all-purpose* 'relational' or *dedicated* 'proprietive'). The QAdj-*like* behaviour of *-ed* Adjs is therefore not a likeness; rather, they are truly converted QAdjs, akin to Latinate RAdjs. This recognition enables the resolution of the idiosyncratic grammatical characteristics of *-ed* Adjs.[23]

As a further remark, as mentioned at the beginning of Section 5, given that RAdjs exhibit undeniable nominal properties in comparison to prototypical QAdjs (see Section 2), RAdj-to-QAdj alternation is, indeed, a conversion rather than a coercion. In this respect, although the relationship between *English* and *very English*, as in (42c), is deemed an *intra*-categorial conversion, it can also be viewed as a *cross*-categorial (or primary word-class) conversion (i.e., N-to-Adj). Conversely, the nominal domain in (42a) and verbal domain in (42b) can be considered genuine instances of secondary word-class conversion, as the former alters referents and the latter modifies argument structures within the respective domains. The adjectival domain has thus been studied in connection with the concept of *mixed categories* (cf. Spencer and Nikolaeva 2017; Nikolaeva and Spencer 2020). More importantly, the same imbalance in scholarly attention is evident not only in *cross*-categorial derivations, as mentioned in the Introduction, but also here in such *intra*-categorial conversions between nouns in (42a), verbs in (42b), and adjectives in (42c) (cf. Nagano 2018a, pp. 185–86).

This situation necessitates a more thorough investigation into the case of the secondary word-class conversion of adjectives.

## 6. Conclusions

In this study, my primary achievement lies in unveiling the adjectival properties inherent to denominal *-ed* Adjs. The three main questions in the Introduction, the answers to which were revealed through this analysis, are repeated as follows:

- What specific properties do *-ed* Adjs intrinsically possess? —The adjectives in question exhibit nominal properties that mirror those of their base nouns, along with distinct proprietive semantics.
- Into which adjectival categories (RAdjs or QAdjs) should *-ed* Adjs be classified? —The relevant adjectives are categorised as a type of RAdj rather than QAdjs due to their nominal properties, which resemble those of general RAdjs. This categorisation is supported by the property inheritance observed in deverbal nominalisations and the semantic nature of proprietiveness. Additionally, it primarily hinges on the etymological features of the suffix, particularly the [±Latinate] feature. General RAdjs bear the [+Latinate] feature, while *-ed* Adjs exhibit the [–Latinate] feature. It is noteworthy that *-ed* Adjs distinctly embody proprietive semantics due to the *dedicated* suffix *-ed*, creating meaning-bearing RAdjs, with their relational meaning intricately interwoven with the classificatory function—a characteristic that stands as prototypical for all RAdjs.
- If *-ed* Adjs are primarily RAdjs, how can their behaviours as QAdjs be analysed and understood? —The adjectives, originally classified as RAdjs, undergo conversion into QAdjs. Similar to Latinate RAdjs, this conversion process can be analysed by introducing the *true* modal operator. However, these adjectives still set themselves apart from Latinate RAdjs in that they represent not only a class-membership relation but also similative semantics due to the effect of proprietive–similative polysemy induced by the *dedicated* suffix.

Despite the limited attention afforded to this particular class of adjectives, *-ed* Adjs, as a subset of RAdjs, present notably intriguing characteristics. In my summarising remarks, however, *-ed* Adjs emerge as a distinctive category within English RAdjs, seamlessly transitioning into genuine QAdjs when imbued with additional qualitative properties.

Finally, a matter that seems highly relevant to this study, and that warrants consideration, is the issue of adjectival passives prefixed with *be-*. As elucidated by Bruening (2014, §7.2), these adjectival passives pose challenges involving specific adjectives, such as *bedecked*, *bedraggled*, *bejewelled*, *belated*, *beloved*, *bemused*, *benighted*, *benumbed*, *bereaved*, *beribboned*, *beringed*, and *besotted*. OED (2023, s.v. *be-7*) also notes that "[s]ome of these words have no form without *be-*, [. . .]". The inherent ambiguity surrounding the bases of these adjectives, some of which seemingly derive from verbs, introduces a noteworthy complexity. However, the *be-* prefix, conventionally construed as *aspectual*, prompts intriguing inquiries regarding its attachment to potential bases. How does it harmonise with bases that may not conform to its typical aspectual function? This intricate issue, which is relevant to our exploration of *-ed* Adjs, demands additional scrutiny and remains earmarked for future research endeavours.

**Funding:** This study was financially supported by the Japan Society for the Promotion Sciences (JSPS) KAKENHI Grant Number JP23K12202.

**Institutional Review Board Statement:** Not applicable.

**Informed Consent Statement:** Not applicable.

**Data Availability Statement:** The original contributions presented in the study are included in the article; further inquiries can be directed to the corresponding author.

**Acknowledgments:** I would like to express my gratitude for the valuable remarks of the academic editors and the four anonymous reviewers, which afforded me the opportunity to reconsider and develop the arguments presented in my first draft. I am also immensely grateful to Akiko Nagano

and Ryohei Naya, as our discussions have significantly enhanced my understanding of this research topic and the related phenomena. Their insights and expertise have proven indispensable in refining my work. Any residual errors are my sole responsibility.

**Conflicts of Interest:** The author declares no conflicts of interest.

## Notes

1.  An anonymous reviewer suggests that the phrase *a large-gardened house* is, in fact, acceptable but is still far less common than *a red-roofed house*. Here, Hudson (1975) seems to judge the examples from the viewpoints of the effect of presupposition (or simply collocational restriction) and inalienable possession. The selectional restriction of *-ed* Adjs has long-interested many scholars, but I will leave this for future research.

2.  In contrast, Ishida and Naya (2022) reexamine the synonymous relation between RAdj-N and N-N, arguing that the range of denotations of RAdj-N combinations is more restricted than that of N-N combinations. The former expresses a *kind* relation (cf. Warren 1984; McNally and Boleda 2004; Shimada and Nagano 2018), whereas the latter conveys a *contextually defined* relation. For example, whereas *allergy illness* (N-N) can express 'an illness consisting of/causing/caused by allergy', *allergic illness* (RAdj-N) can only express 'an illness caused by allergy' (see Ishida and Naya 2022, p. 23 for other examples). The reason why RAdj-N denotes a kind relation is rooted in the inherent classificatory function of RAdjs and their derivational process (see Shimada and Nagano 2018 for their structural analysis). Ishida and Naya (2022) consequently propose that RAdj-N can be semantically classified as a subset of N-N.

3.  In fact, Nagano (2013, 2015) includes *-ed* Adjs (e.g., *bearded*) within her analysis of RAdjs; however, she does not provide any explanation or definitions for this category.

4.  Morita (2024, pp. 192–94) argues that *-ed* Adjs are phrase-derivative (cf. phrasal compound) and that their unsaturated and inalienable semantic nature is the crucial factor that allows for the incorporation of NPs (e.g., [A N]_{NP}) into *-ed* adjectivalisation. This is in contrast to suffixes like *-y* and *-ful*, which cannot incorporate phrases due to their semantically saturated nature (i.e., 'having the unusual (amount of)~'), such as (*long-)*hairy*, (*big-)*chesty*, and (*strong-)*wilful*. However, *-ed* Adjs generally preclude attachment to phrases, as shown in the following examples: *\*John is [more blue than the sea]-eyed* (Tsujioka 2002, p. 164) and *\*[blue eye I admire]'d boy*. This outcome is predictable, given that "most suffixes do not attach freely to phrases, but only to roots or words" (Plag 2018, p. 151). This seemingly confusing situation can be explained by genuine pragmatics (what Nishiyama (2017, p. 163) calls a *cliché*). This phenomenon is reminiscent of the following contrast between lexicalised phrases and genuine syntactic phrases: [*American history*] *teacher* vs. \*[*recent history*] *teacher* (Bresnan and Mchombo 1995, p. 193); [*open door*] *policy* vs. \*[*wooden door*] *policy* (Carstairs-McCarthy 2018, pp. 88–89). Thus, unless the input phrases are lexicalised, they may be excluded from *-ed* suffixation. Incidentally, the edge clitic *'s*, functioning as a possessive morpheme (cf. Miller 1991; see also Barker 2011 for other examples and discussions), can attach to the final word at the rightmost edge of a full determiner phrase (e.g., [*the boy over there*]*'s blue eye*).

5.  As observed by an anonymous reviewer, examples such as *boxed*, *canned*, or *tinned* do not seem to simply mean possessional but rather the concept of 'container'. For example, *boxed* does not literally mean 'with a box'; rather, it denotes 'enclosed in a box' (OED 2023, s.v. *boxed adjective₂*). However, I argue that these instances can be considered a subtype of *-ed* Adjs, as they are simply associated with a general cognitive model. Specifically, a conceptual metaphor, POSSESSION IS LOCATION, is crucially involved here. As explained by Lyons (1977, p. 474), possessional expressions "are to be regarded as a subclass of locatives". In other words, possessionals and locatives share an abstract conceptualisation. Therefore, even though the derivatives are all semantically locative and their grammatical form is possessional (i.e., *-ed*), they can be grouped in a similar manner as other *-ed* Adjs. In this regard, *wooded* 'covered with trees' can also be a type of this. Incidentally, some derived nouns can also be the input of *-ed* Adjs, such as *sour-visaged* (*-age*), *good-appearanced* (*-ance*)/*average-intelligenced* (*-ence*), *low-ceilinged* (*-ing*), *fair-complexioned* (*-ion*), *battlemented* (*-ment*), *average-lengthed* (*-th*) (Takehisa 2017, p. 197).

6.  The term *bahuvrihi* (*bahu-vrihi-*: lit. 'much rice') is a Sanskrit word signifying 'having much rice'. A bahuvrihi compound denotes a referent by articulating a particular characteristic or quality that the referent possesses. And it is important to note that it is exocentric, meaning that the compound does not function as a hyponym of the head. An example of this can be seen in the case of a *sabretooth* (i.e., smilodon), which is neither a sabre nor a tooth, but rather a feline possessing teeth that resemble a sabre.

7.  A cranberry morpheme (or a cran-morph) is a category of bound morpheme that lacks the ability to be assigned an autonomous meaning and grammatical function. Nevertheless, it still functions to differentiate one word from another. For example, *-berry* of *cranberry* is evidently a free morpheme, but the initial element *cran-* does not seem to occur in other words and lacks an independent meaning. Other typical examples are the *huckle-* of *huckleberry* and *rasp-* of *raspberry* (Lieber 2022, pp. 46–47). In addition, Brown and Miller (2019, p. 115) raise the *kith* of *kith and kin* and *-ric* of *bishopric* as possible examples.

8.  Bound stems are identified as bases incapable of autonomous word-forms (Haspelmath and Sims 2010, p. 21). The following question then arises: how can these serve as inputs for RAdj formation? I posit that they undergo a (strong) suppletion akin to the phenomenon observed in pairs such as *go/went* and *good/better* (cf. Haspelmath and Sims 2010, p. 25; see also Rudes 1980; Koshiishi 2002, 2011; Hogg 2003; Maiden 2004; Corbett 2007; Veselinova 2006, 2020; Maiden and Thornton 2022). Expanding on Nagano's (2013, 2015) analysis of RAdjs as representing morphological, realisational variations in prepositional phrases (hereafter

PP) (e.g., *bearded* is a realisational form of *with beard*; *dental* is a suppletive form derived from the corresponding PP, *with teeth*), I argue that the underlying structure can be the corresponding PP on the right side, as exemplified in (i) (see (5) and (6) again, as well as Nagano (2013) for the original analysis and detailed discussion).

| (i) | | | | |
|---|---|---|---|---|
| | a. | *naked* legs | a′. | legs *with the state of being nude / in the nude* |
| | b. | *wicked* uncle | b′. | uncle *with a witch/wizard-like character* |
| | c. | *blackavised* chap | c′. | chap *with a black-face/dark-complexion* |
| | d. | *jagged* edge | d′. | edge *with a jag / dag / tag* |
| | e. | *fructed* tree | e′. | tree *with fruit* |

The analysis of each example in (i) is based on the respective entries in OED (2023). Regarding *naked* in (ia) first, the explanation reveals its inheritance from various Germanic languages (e.g., Old Frisian, Middle Dutch, Old Icelandic, and so forth). With various word forms in different languages, and with the same Indo-European base, *nude* is borrowed from the classical Latin *nūdus*, but the explanation of the forms is still not clear. This fact at least indicates that *naked* is no longer analysable as a concatenative formation of the base *nake-* and the suffix *-ed*, suggesting that *naked* may either be borrowed from other languages or represent a suppletive form of its abstract corresponding PP, *with the state of being nude / in the nude*, as indicated in (ia′). While evidence is limited, the latter analysis appears more suitable based on OED's (2023) explanation. Adopting suppletion allows for the derivation of 'missing inputs' such as *naked* (~*nude*), aligning with other suppletive (or often called *collateral*, Koshiishi 2002, 2011) RAdjs such as *dental* (~*teeth*), *canine* (~*dog*), and *vernal* (~*spring*).As for *wicked* in (ib), although it seemingly derives from *wretched*, within English, the etymology of the base *wick* is either (I) formed by conversion or (II) altered by another lexical item meaning 'witch or wizard', such as *wicke*/*wikke* (different forms of *wicca* in Old English and *wicci* in Middle English). With little evidence provided by OED (2023), neither possibility should be discarded, suggesting that the latter possibility is straightforwardly associated with our suppletion analysis.*Blackavised* in (ic) is intriguing yet complex. The English base *black* and the French *vis* 'face' are likely combined with the English euphonic *-a-*. While *vice* in English is the corresponding word for *vis*, it has long been obsolete. As a formal fit, the compound adjective *black+advised* is conceivable, but its original sense is, in fact, 'having bad intentions'. Compound adjectives such as *dark-avised* or *light-avised* are modelled on *blackavised* considerably later. Hence, due to the etymological absence, *blackavised* also represents a form of strong suppletion, an alternative realisation of the PPs as depicted in (ie′), for instance.*Jagged* in (id) appears more complex and distinct from other examples. Its supposed base *jag* 'one of the dags or pendants made by cutting the edge of a garment' is described as "[p]robably an imitative or expressive formation" (OED 2023, s.v. *jag noun₁*) and coincides with *dag* 'a loose end; a dangling shred' (Webster's 1913, s.v. *dag noun*) or *tag* 'something slight hanging loosely' (Webster's 1913, s.v. *tag noun₁*) (or *rag*) in some senses, though both origins are uncertain. Therefore, *jag* may be copied by its phonological counterpart *dag/tag*, and *jagged* is derived from the imitative word *jag*, possibly through a unique suppletion process. Additionally, it is noteworthy that both *tagged* and (potentially) *dagged* can exhibit identical phonology to *jagged*, represented as /ɪd/. The *imitative* or *expressive* nature of *jag* may render its lexical status unstable, often described as synchronically unproductive and uncommon (cf. Allen 1978; Harley 2006).Lastly, the base of *fructed* in (ie) is borrowed from the Latin *frūctus* 'fruit', with no corresponding nominal in English, representing a typical case of suppletion.The considerations derived from the suppletion analysis of *-ed* Adjs offer a renewed examination of the inherent nature of suppletion in English, spanning not only diachronic but also synchronic perspectives (Hogg 2003; see also Naya and Ishida 2023 for the application of Nagano's (2013, 2015) analysis regarding RAdjs originating from 'imaginary' dvandva compounds in English).

9   Academic editors further point out that many proprietive *-ed* Adjs have a CARITIVE counterpart, meaning 'lacking N, without N' (e.g., Nikolaeva and Spencer 2020, p. 105), as paraphrased with the suffix *-less*, suggesting that *naked* may be a suppletive form of *clothes-less*. This perspective seems to partially align with my initial analysis, as illustrated in Note 8.

10  An anonymous reviewer observes that deverbal *-ed* Adjs can be markedly written as *-èd*, such as *blessèd* in psalms and *learnèd* in *learnèd suffix*. Furthermore, another reviewer proposes that a more comprehensive analysis of the stress placement of *-ed* Adjs, as discussed by Kiparsky (1982), amongst others, is warranted. However, this aspect will be the focus of future research.

11  Although Bauer et al. (2013, p. 313) did not provide the definition of the semantics of the suffix *-ed*, nor touch on *proprietive*, they explain that the suffix expresses an *ornative* sense. On the basis of their analysis, the distinction between ornative and proprietive should be attributed to either the suffix itself or the derivative as a whole. Therefore, *ornative* is specific to the *-ed* suffix, while proprietive semantics is associated with the derivative, *-ed* Adjs. In light of the ensuing discussions on possessive–proprietive asymmetry in Section 3.3 and proprietive–similative polysemy in Section 5.3, I have decided to employ *proprietive* in the present study.

12  An anonymous reviewer claims that a phrase such as *a very over-fished lake* sounds acceptable, and *fished* in this case may not be totally a non-scaler but rather be a lower closed scale. In either case, *-ed* Adjs, compared to *-y* Adjs, are considered to be the closed-scale type. Moreover, the prefix *over-* of the phrase here can contribute to varying the scale type.

13  An anonymous reviewer correctly points out that, due to the open-scale nature of *-y* Adjs and closed-scale *-ed* Adjs, *completely clouded* can be more acceptable than *completely cloudy*.

14  Marchand (1969, p. 270) and Allen (1978, pp. 175–76) note that *-en* Adjs also express a similative meaning, as exemplified in *a silken voice*, *flaxen hair*, and *a leaden smile*.

[15] Notably, although an anonymous reviewer contends that (ii) is completely grammatically acceptable, Koenig and Launer (1997, p. 63) clearly state that *bearded* in a predicate position as in (ii) is "of limited acceptability, and many consider it to be somewhat archaic", providing the more commonly understood context in (iii).

(ii)      ?The man was bearded.      (A beard is present on the man.)

(iii)      A:    Is this the man you saw robbing the store?
             B:    No, the man I saw was bearded.

In this scenario, the speaker endeavours to describe the criminal situation, specifically identifying the man involved in the robbery as having a beard. Koenig and Launer (1997, p. 63) further provide other types of adjectives that denote composition as in (iva) and nationality as in (ivb).

(iv)      a.    ?The glove is woollen.      (The glove is made of wool.)
            b.    ?The vodka is Russian.      (The vodka is made in Russia.)

Generally, these adjectives are also not acceptable in a predicate position, particularly within the context of formal discourse, albeit that they may be somewhat tolerated in colloquial speech.

[16] With respect to this example, an anonymous reviewer has enigmatically suggested that the reason it may not be entirely effective is potentially due to the fact that the grammatical subject, *John*, is not an indefinite noun, unlike the other two examples. However, the attributive form, such as *a true foul-mouthed John* 'a man named John who can truly be called foul-mouthed', is deemed acceptable. In light of this observation, it is plausible that definiteness may play a role here.

[17] Notably, the quantifiability criterion in (3e) (and repeated in (19e)) was initially proposed by Levi (1978, p. 24) and has been effectively adopted by Fábregas (2007, p. 8) and Moreno (2018, p. 58) as a benchmark for Spanish RAdjs.

[18] I thank academic editors for helping me address this point.

[19] Interestingly, Naya (2016) points out, based on historical data from the *Oxford English Dictionary*, that the *-ment* nominalisations of ENs and RNs are not diachronically but rather *independently* derived from each other. This can be a rare exceptional case in Nagano's (2015) sense and poses a challenge to my argument regarding the derivation process from RAdjs to QAdjs.

[20] I express gratitude to academic editors for the constructive suggestion to discuss this point, which aided in refining the argument presented.

[21] Hoeksema (1985, p. 179), however, vehemently questions this type of analysis, suggesting that, even if the input is undoubtedly regarded as a relational noun, there are examples that are deemed unacceptable, such as *threesistered*, *cruelfathered*, and *manyfriended* in both English and Dutch.

[22] Incidentally, *-en* and *-ern* Adjs, as Germanic RAdjs exemplified in (v), can be analysed in a similar manner.

(v)    a.    His face was *wooden*, like the carved figurehead of a ship, and it gave no more sign of what went on inside him than the ship's figurehead gives of the storms at sea.

(From *The Heart is a Lonely Hunter* by Carson McCullers)

      b.    Her voice was *southern*, with a softness like the rustle of magnolia leaves, and a cadence that reminded him of the slow, meandering rivers of his youth.

(From *The Optimist's Daughter* by Eudora Welty)

In (va), *wooden* metaphorically describes the lack of expression or emotion on the subject's *face*, creating an image of stiffness and rigidity. In (vb), *southern* emphasises the distinctive regional qualities of the subject's *voice*. These examples similarly represent metaphorical class membership through inferences derived from the semantics of the *true* operator.

[23] Incidentally, although it may be speculative, the absence of an independent lexical status for certain N-*ed* forms such as *wifed*, *doored*, or *carred* (beyond the semantico-pragmatic factors given in the literature) may now be attributed to three relevant factors: (i) possessive–proprietive asymmetry; (ii) blocking by other possessive constructions; and (iii) no affixal rivalry with *-y*. Concerning (i), for instance, compared to the pair *tattooed arm* vs. *arm tattoo*, *wheeled transport* is acceptable, but $^?$*transport wheel* 'wheel for transport vehicles' is not (Nikolaeva and Spencer 2020, p. 102). In terms of (ii), other (pre- or postnominal) possessive constructions, such as *the man's wife* or *the wife of the man*, seem to block the possible corresponding proprietive constructions, *a wifed man* (cf. *the lady's car/the car of the lady* vs. *a carred lady*). The last point, indicated by (iii), presupposes that unacceptable N-*ed* Adjs also appear to lack affixal rivalry with *-y* Adjs (i.e., *wifey*, *doory*, *carry* (cf. *carry* < *carr*$_N$); however, there are exceptional cases in this regard (*husbanded* vs. *husbandy*).

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
