# Peer review of "Denominal -ed Adjectives and Their Adjectival Status in English Morphology"

_languages, doi:10.3390/languages9050169_

Round 1

Reviewer 1 Report

Comments and Suggestions for Authors

This is a very interesting paper, embedded in and building on the literature related to this subject, presenting several new insights.

Some small remarks:

p. 2: The distinction between the two types of deverbal nouns cannot be fully compared to the distinction between RAdjs and QAdjs. Whereas the RAdjs and QAdjs in (1) are presented as two distinct types of adjectives, event nominals and result nominals are related. In (2) the same nouns present two different interpretations.

p. 3: Diachronically 'visaged' and 'ceilinged' may be derived, but from a synchronic point of view this may be debated.

p. 4, Some morphological notions could be better explained, such as 'bahuvrihi' and 'cranberry-morpheme'.

p. 4: What could be added, is that deverbal -ed adjectives may also be written as -èd: blessèd (in psalms), learnèd (in learnèd suffix)

p. 5: the distinction between 'possessive' and 'proprietive' is not sufficiently well explained.

p. 5: It is not clear why common nouns are called non-referential and proper nouns referential.

p. 5-6: It is unclear what is meant in the lines 241-256. 'Riverbank' and 'campsite' are nouns and not adjectives. Both are called PROP.adj. There is no illustration of an POSS. Adj. although the description suggests that (7a) would rather have the POSS.Adj interpretation. It is not clear what these lines want to show.

p. 6, line 306. Shouldn't 'nominal' be 'adjectival'?

p. 416, line 416: The word 'other' suggests that -ed Adjs are QAdjs, but we learn only later that they are analyzed as RAdjs that may have a QAdj's behaviour and interpretation.

p. 15: The goal of adding the lines 751-757 is not clear. It is not clear what they mean and how they contribute to the argumentation.

p. 21:  p. 1079-1080: does this mean that -ful and -ish are analyzed as inflectional suffixes?

Some typos:

p. 14, line 711: prviding --> providing

p. 15, line 751: expect --> respect

p. 20, line 1025: a true --> is a true

p. 23, line 1184: realsational --> realisational

p. 24, line 1208: primarycharacteristic --> primary characteristic

p. 26, fn. 1: exmeplifeid --> exemplified

line 225: having blue-eyes, line 446: blue-eye, line 1067: with blue-eye. 

Comments on the Quality of English Language

English is very good. There are some small typos.

Reviewer 2 Report

Comments and Suggestions for Authors

In general, the author(s) do a quite good job providing an argument with ample evidence of how to assess and categorize -ed adjectivals in English. While I do think there needs to be a bit more set-up to characterize the novelty and contributions of this work (see line-item comments below), I appreciate how much this study builds on the recent work of other morphologists to argue for -ed ADJs as a group distinct from the Latinate ADJs. I found the discussion of differences in etymological features, the relevance of suppletion, the idea of "proprietive" relations, and many more concepts interesting and likely to generate interest and further study among word-formation scholars. While some readers might think there is perhaps too much reliance on Nagano's work (and perhaps a few spots where they could pare down those references), I appreciated the authors' thoughtful engagement with Nagano's framework and integration/application of it alongside the work of other scholars such as Ishida & Naya and Spencer & Nikolaeva. This felt like a study that moved all of these scholars' work forward while adding new insights on -ed adjectives as a distinct group.

Even so, there were a number of spots that needed a bit of clarification or elaboration. So I'm recommending acceptance of this article with some minor revisions. I'll list some suggestions for additions, revisions, and clarifications below (with line numbers in the article indicated with ll.):

l. 15 (in the abstract, but also in section 1): It likely won't be surprising to readers that "-ed Adjs manifest as the inflected word-type" since -ed is so often characterized as an inflection. So can the abstract and intro set up a bit more why this is actually a less obvious or intuitive finding than it may seem on the surface? I very much understand this paper's argument and contribution to this question by the paper's end, but I wasn't as clear about it in the early stages and the abstract. In other words, why wouldn't we assume to begin with that it was an inflected word-type?

ll. 41-42: Add a sentence or two here explaining the difference between "qualifying" and "classifying" (and confirm whether previous studies have shown this is a clear-cut distinction or if there are any edge cases where denominal suffixes might be seen as either or both qualifying and classifying).

l. 68: "categorial status as 'verbs'" --> ? Do the authors mean something like "as 'verb'-derived" (in terms of semantics?). Need to make this clearer, since they aren't verbs themselves.

ll. 68-70: Provide contextual examples with a couple of these Adjs from 1a and 1b to illustrate the contrast, perhaps parallel to what (2) and (3) are doing with the nominals.

l. 74: Should maybe also cite Kaunisto's extensive work on -ic/-ical (e.g., Kaunisto, Mark. 2007. Variation and Change in the Lexicon: a Corpus-Based Analysis of Adjectives in English Ending in -ic and -ical.)

ll. 85-86: Is there evidence that the asterisked forms are all ungrammatical? Is the claim here about absolute unacceptability, or less likelihood of use? (Just in terms of frequency?) Personally I'd accept "large-gardened," for example, but I have a sense that it's far less common than "red-roofed.")

l. 126: Do the authors agree with this very strong claim from the OED that -ed can be attached "without restriction to any noun"? I'm not even sure if this paper's examples support that idea fully (or if there's any suffix in existence that attaches to anything without restrictions of some sort!)

ll, 248-251: Add a sentence to explain how both 7a and 7b are Prop.Adj? (Having a bit of trouble esp. with 7b, and how campsite is possessor and riverbank is interpreted as possession).

l. 288: I think the closed vs open distinction for -ed vs. -y is right, but I'd personally accept something like "a very over-fished lake." So fished is maybe not totally non-scalar (would that be "lower closed scale"?)

 l. 297: I think "completely cloudy" would be passable, but I agree with the implication that "completely clouded" is much more acceptable. Maybe reframe a bit here and talk about relative acceptability: "due to the open-scale nature of -y and closed-scale nature of -ed, completely clouded may be more acceptable than completely cloudy").

l. 351+: I don't know if there's been a change since the 90s, but this use of bearded in a predicate adjective position is completely grammatically acceptable for me. Maybe delete this example and some of the discussion below and just provide examples 14b and 14c?

ll. 449-450: Spell this part out a bit more--there's a lot of information in this and the prior section, so it's easy to lose track. What aligns with what, specifically?

l. 554: I think "the" is needed before "nuclear one".

l. 622: I stumbled a bit on 16a. This one doesn't work for me. I wonder if it's because John isn't an indefinite noun as in the other two examples? I think "a true foul-mouthed John" = 'a man named John who can truly be called foul-mouthed' works, though. (I'm not sure why definiteness or indefiniteness would matter here, but that's just my reaction to that one example.)

l. 751: "expect" doesn't make sense here. Do the authors mean "respect"?

ll. 755-757: I don't have any concerns about use of Nagano's framework, and if anything think its application in this article is a major plus. But this is the sort of spot where the author(s) could pare down a bit of the reference/praise perhaps since it's frequently invoked. (And it's worth going back through the paper to see if there's anywhere else to pare back a bit.)

ll. 803-805: Can the authors use a different notation in 43a-c so that the dashes don't get interpreted as minus signs (e.g. you don't want this to read as FRONT minus ADVERB, right)? Maybe a short hyphen or slash would work better here? (It's only an issue because + appears as well.)

l. 814: "of" needs to be "if," maybe?

ll. 855-857: Could the authors explain why a plural marker would be "expected" to appear before -ed? It could be worth considering this in light of Stump's (1998: 14-19) discussion of inflection as closure: that it's unusual for inflections to be closer to the root than derivative suffixes (but perhaps -ed having inflectional as well as derivational tendencies is relevant here. In other words, is plural-s disallowed because the derivational -ed is applied "first")?

See: Stump, Gregory. 1998. “Inflection.” The Handbook of Morphology, eds.Andrew Spencer and Arnold Zwicky, 13-43.

ll. 934-936: Can the authors gloss dagged/tagged and dag/tag? (tag/tagged will be more familiar to readers, but dag/dagged much less so.)

ll. 1170-1171: The description here seems different from Figure 1: the figure seems to indicate that RAdj-N is a subclass of N-N, but the description sounds like RAdj-N is encompassing the N-N (rather than vice versa). Can the authors make how to read this figure clearer? (I may be misinterpreting the figure or the references to "latter" and "former" in this section, but the authors need to make sure the figure and the prose description are in complete alignment.)

Comments on the Quality of English Language

It's quite clear, mostly, but there are several errors in word choice or missing words. I didn't comment on them all, but I do think the editors need to advise the authors to proofread for typos.

Reviewer 3 Report

Comments and Suggestions for Authors

The aim of this paper is to analyze de nominal -ed adjectives in English, looking at their phonological, semantic, morphological and lexical properties and using the Lexeme-Morpheme Base Morphology. The paper fits the aim of Special Issue in that it discusses theoretically framed questions about word-formation found in English. In addition, the main contribution of this paper is that it successfully describes the adjectival/nominal status of these words by using the previous literature’s ways. 

Thus, I support for the acceptance of the manuscript for Special Issues, Languages 2023 as long as the set of points raised here are carefully reviewed. 

1. Overall

The questions in Section 1 should be answered in either Conclusion or the later sections. The author(s) seem to, but it is not obvious, so please state which questions are answered in each section or part of the paper. The organization of the paper should be reconsidered in this way. 

2. Questions about the data

1) -ed adjectives in English: what about the words like boxed, canned, tinned, which probably mean ‘container’? 

2) The data in (4) a - e: For the basic properties of -ed adjectives, it would be easier for the readers to see the analyses of the phonological, semantic, and morphosyntactic properties for all the data set, probably there are some patterns for (4a), for example? 

3) (4) a- e: Lines 161 - 173, the author(s) writes the descriptions about the (4b -d), but there are no explanations about (a) and (e). What are the differences between them, for example? In addition, how can b-d be bahuvrihi compounds? Define and explain. Does it mean that the category of the whole words is different from the second constituent? 

3. Analyses of the words

1) Section 2.2, the author(s) discuss the phonological analysis of the words. There must be more change of placement of the stress. There is work on placement of stress - e.g. Lexical Phonology by Kiparsky, for example. 

2) Section 2.3

A. Can the author(s) please distinguish between Proprietive and ornative? How are they different and how do you know which ones are which? Expand the paragraph of 220-228. 

B. Loose association - this is very “loose” definition. Give us more examples, please. 

C. 229 - 240 - Explain what Nagano (2023) does for ‘part-whole’ and the other two relationships, and maybe relate these to your data (4a-e). 

D. (8a-f)

Are they from (4a-e)? How are they related to each other? 

E. Overall, in this Section, the author(s) are trying to see if -ed adjectives are semantically similar to RA or QA. Then, why not first discuss what QA and RA in the beginning of this section for the readers to see the properties more clearly? I am confused that in this section, the author(s) seem to say that -ed adjectives are not like QA, but in the later section, they ARE like QA.

3) The same argument goes for the following section. First, discuss what are the properties of QA, compared to RA and argue if -ed adjectives are similar to either of them. Are the examples (19) necessary? I do not see the point of these examples. In addition, the examples (22) seem to be about Lexical Integrity test. Maybe expand on this? 

4) At the end of Section 2, the author(s) say that there are problems. What are the problems? Explain more. 

4. Section 3 

Lexeme-Morpheme Base Morphology analyses - (29), and all the other analyses, explain more clearly about them and then, discuss. Maybe at the beginning of Section3, it is clearer if LMBM is explained. Section 4.1 with Nagano 2013, 2015) and Section 4.2 with  Naha & Ishida (2023) should be moved to Section 3, For example. The explanations about LMBM are scattered in different sections, which could confuse the readers. 

 5. Other points 

1) Please check the References - e.g. Nagano (2023) is not there. 

Comments on the Quality of English Language

There are some spelling mistakes. Please check English Language again. 

Reviewer 4 Report

Comments and Suggestions for Authors

This is a clearly written article, which very clearly addresses the nature of -ed adjectives, based on the initial distinction between Qualitative and Relational Adjectives, the former being more related to adjectives, the latter to nouns. The author considers all possible angles for the inspection of -ed Adjs and is well aware of the previously existing proposals and viewpoints on the topic. Overall, after considering the possible analyses and the features considered in the study, I agree that the most logical position for -ed Adjs lies between Qualitative and Relational Adjs.

The view of the phenomenon in the light of RAdj-to-QAdj conversion is a surprising and challenging one. This position is defended with the example of “nuclear”, in turn adopted from Bauer et al. (2013). For me, the fundamental assertion is that the intricacies in this matter are caused by “the dual-standard suffixal status of -ed in Present-day English, encompassing both N-ed and V-ed”. I suggest adding an additional couple of examples in this section in order to facilitate an already complex topic.

Data

At a given point, the author cites the OED Online as the source of some of the units. The OED considers various possible stagers for a word’s usage status. I would like to ask if the status of all the units in the study is “in use”, or if any unit is regarded as marginal, obsolete, etc. Whichever the answer, this piece of information should be inserted in the paper when the OED is mentioned.

Minor remarks:

·        Page 4. Is the remark on bahuvrihi meaning ‘having much rice’ helpful in this context? If not, please remove.

·        Page 14: there is a typo “prviding”.

·     Pages 20-21: mention is made of languages other than English, and in particular there is an example from Udihe. I wonder if this is relevant to the scope of the paper, which I had assumed targets exclusively English adjectives. I would make this clear from the beginning of the article, since the Udihe example is a bit surprising at this point of the work.

Round 2

Reviewer 3 Report

Comments and Suggestions for Authors

The aim of this paper is to analyze de nominal -ed adjectives in English, looking at their phonological, semantic, morphological and lexical properties. The paper fits the aim of Special Issue in that it discusses theoretically framed questions about word-formation found in English. In addition, the main contribution of this paper is that it successfully describes the adjectival/nominal status of these words by using the previous literature’s ways. 

Thus, I support for the acceptance of the manuscript for Special Issues, Languages 2023 for the following reasons. 

1) Overall

The readability of the paper has improved greatly, because the structure of the paper has been changed since the previous version of the paper. The research questions and the answers are all related to each other with the data provided through the paper. Even the reader is not specialized in the topic of the research, they can follow the content very easily.

2)  Is the content succinctly described and contextualized with respect to previous and present theoretical background and empirical research (if applicable) on the topic?

Yes, Introduction gives the overall research background to the previous research on denominal adjectives and the author has clear research questions based on them. 

3) Are all the cited references relevant to the research?

The paper explains and discusses the previous literature very clearly and then, expands the arguments based no the previous literature’s arguments as well as the author’s. 

4) Are the research design, questions, hypotheses and methods clearly stated? 

Yes, the same as 2) and the author’s methods are also stated (the data is from the previous literature). 

5) Are the arguments and discussion of findings coherent, balanced and compelling?

Yes, the author’s findings (semantic, phonological and morphological properties of -ed adjectives) are all clearly stated in relation to the properties of other types of denominal adjectives, which have been discussed in the previous literature. In addition, it is also intriguing to see the differences between Latinate -ed and non-Latinate suffixes. 

6) For empirical research... 

Yes. The data is very easy to follow.  

7) References 

Yes. I have checked the main references. They are all appropriate references for this topic of research.  

8) Conclusions 

Yes. Introduction and conclusions are clearly connected to each other. 

Thus, I support the publication of the paper. 

Author Response

I am grateful for your examination and completely positive feedback of the initially revised version of my manuscript.